# ON THE EXISTENCE OF UNIVERSAL LOTTERY TICKETS

**Rebekka Burkholz**
CISPA Helmholtz Center
for Information Security
burkholz@cispa.de

**Nilanjana Laha, Rajarshi Mukherjee**
Harvard T.H. Chan School of Public Health
rmukherj@hsph.harvard.edu

**Alkis Gotovos**
MIT CSAIL
alkisg@mit.edu

## ABSTRACT

The lottery ticket hypothesis conjectures the existence of sparse subnetworks of large randomly initialized deep neural networks that can be successfully trained in isolation. Recent work has experimentally observed that some of these tickets can be practically reused across a variety of tasks, hinting at some form of universality. We formalize this concept and theoretically prove that not only do such universal tickets exist but they also do not require further training. Our proofs introduce a couple of technical innovations related to pruning for strong lottery tickets, including extensions of subset sum results and a strategy to leverage higher amounts of depth. Our explicit sparse constructions of universal function families might be of independent interest, as they highlight representational benefits induced by univariate convolutional architectures.

## 1 INTRODUCTION

Deep learning has achieved major breakthroughs in a variety of tasks (LeCun et al., 1990; Schmidhuber, 2015), yet, it comes at a considerable computational cost (Sharir et al., 2020), which is exaggerated by the recent trend towards ever wider and deeper neural network architectures. Reducing the size of the networks before training could therefore significantly broaden the applicability of deep learning, lower its environmental impact, and increase access (Dhar, 2020). However, such sparse representations are often difficult to learn, as they may not enjoy the benefits associated with over-parameterization (Belkin et al., 2019).

Frankle & Carbin (2019) provided a proof of concept that sparse neural network architectures are well trainable if initialized appropriately. Their lottery ticket hypothesis states that *a randomly-initialized network contains a small subnetwork that can compete with the performance of the original network when trained in isolation*. Further, Ramanujan et al. (2020) conjectured the existence of strong lottery tickets, which do not need any further training and achieve competitive performance at their initial parameters. These tickets could thus be obtained by pruning a large randomly initialized deep neural network. Unfortunately, existing pruning algorithms that search for (strong) lottery tickets have high computational demands, which are often comparable to or higher than training the original large network. However, Morcos et al. (2019) posited the existence of so-called *universal* lottery tickets that, once identified, can be effectively reused across a variety of settings.

**Contributions.**

- In this paper, we formalize the notion of universality, and prove a strong version of the original universal lottery ticket conjecture. Namely, we show that a sufficiently over-parameterized, randomly initialized neural network contains a subnetwork that qualifies as a universal lottery ticket without further training of its parameters. Furthermore, it is adapted to a new task only by a linear transformation of its output. This view can explain some empirical observations regarding the required size of universal lottery tickets.

- Our proof relies on the explicit construction of basis functions, for which we find sparse neural network representations that benefit from parameter sharing, as it is realized by convolutional neural networks. The fact that these representations are sparse and universal is the most remarkable insight.

- To show that they can also be obtained by pruning a larger randomly initialized neural network, we extend existing subset sum results (Lueker, 1998) and develop a proof strategy, which might be of independent interest, as it improves current bounds on pruning for general architectures by making the bounds depth dependent. Accordingly, the width of the large random network can scale as $n_0 \geq O(n_t L_t / L_0 \log(n_t L_0 / (L_t \epsilon)))$ to achieve a maximum error $\epsilon$, where $L_t$ denotes the depth and $n_t$ the width of the target network, and $L_0$ the depth of the large network.

- In support of our existence proofs, we adapt standard parameter initialization techniques to a specific non-zero bias initialization and show in experiments that pruning is feasible in the proposed setting under realistic conditions and for different tasks.

**Related work.** The lottery ticket hypothesis (Frankle & Carbin, 2019) and its strong version (Ramanujan et al., 2020) have inspired the proposal of a number of pruning algorithms that either prune before (Wang et al., 2020; Lee et al., 2019; Tanaka et al., 2020) or during and after training (Frankle & Carbin, 2019; Savarese et al., 2020). Usually, they try to find lottery tickets in the weak sense, with the exception of the edge-popup algorithm (Ramanujan et al., 2020) that identifies strong lottery tickets, albeit at less extreme sparsity. In general, network compression is a problem that has been studied for a long time and for good reasons, see, e.g., Lin et al. (2020) for a recent literature discussion. Here we focus specifically on lottery tickets, whose existence has been proven in the strong sense, thus, they can be derived from sufficiently large, randomly initialized deep neural networks by pruning alone. To obtain these results, recent work has also provided lower bounds for the required width of the large randomly initialized neural network (Malach et al., 2020; Pensia et al., 2020; Orseau et al., 2020; Fischer & Burkholz, 2021; 2022). In addition, it was shown that multiple candidate tickets exist that are also robust to parameter quantization (Diffenderfer & Kailkhura, 2021). The significant computational cost associated with finding good lottery tickets has motivated the quest for universal tickets that can be transferred to different tasks (Morcos et al., 2019; Chen et al., 2020). We prove here their existence.

## 1.1 NOTATION

For any $d$-dimensional input $\boldsymbol{x} = (x_1, \ldots, x_d)^T$, let $f(\boldsymbol{x})$ be a fully connected deep neural network with architecture $\bar{n} = [n_0, n_1, ..., n_L]$, i.e., depth $L$ and widths $n_l$ for layers $l = 0, ..., L$, with ReLU activation function $\phi(x) := \max(x, 0)$. An input vector $\boldsymbol{x}^{(0)}$ is mapped to neurons $x_i^{(l)}$ as:

$$\boldsymbol{x}^{(l)} = \phi\left(\boldsymbol{h}^{(l)}\right), \quad \boldsymbol{h}^{(l)} = \boldsymbol{W}^{(l)}\boldsymbol{x}^{(l-1)} + \boldsymbol{b}^{(l)}, \quad \boldsymbol{W}^{(l)} \in \mathbb{R}^{n_{l-1} \times n_l}, \; \boldsymbol{b}^{(l)} \in \mathbb{R}^{n_l}, \tag{1}$$

where $\boldsymbol{h}_i^{(l)}$ is called the pre-activation of neuron $i$, $\boldsymbol{W}^{(l)}$ the weight matrix, and $\boldsymbol{b}^{(l)}$ the bias vector of layer $l$. We also write $\boldsymbol{\theta}$ for the collection of all parameters $\boldsymbol{\theta} := \left(\left(\boldsymbol{W}^{(l)}, \boldsymbol{b}^{(l)}\right)\right)_{l=1}^{L}$ and indicate a dependence of $f$ on the parameters by $f(x|\boldsymbol{\theta})$.

We also use 1-dimensional convolutional layers, for which width $n_l$ refers to the number of channels in architecture $\bar{n}$. For simplicity, we only consider 1-dimensional kernels with stride 1. Larger kernels could simply be pruned to that size and higher strides could be supported as they are defined so that filters overlap. The purpose of such convolutional layers is to represent a univariate function, which is applied to each input component.

Typically, we distinguish three different networks: 1) a large (usually untrained) deep neural network $f_0$, which we also call the mother network, 2) a smaller target network $f$, and 3) a close approximation, our lottery ticket (LT) $f_\epsilon$, which will correspond to a subnetwork of $f_0$. $f_\epsilon$ is obtained by pruning $f_0$, as indicated by a binary mask $\boldsymbol{B} = (b_i)_{i \in \{0,1\}^{|\boldsymbol{\theta}_0|}}$ that specifies for each parameter $\theta_i = b_i \theta_{i,0}$ whether it is set to zero ($b_i = 0$) or inherits the parameter of $f_0$ by $\theta_i = \theta_{i,0}$ (for $b_i = 1$).

We usually provide approximation results with respect to the l1-norm $\|\boldsymbol{x}\| := \sum_i |x_i|$ but they hold for any $p$-norm with $p \geq 1$. $C$ generally stands for a universal constant that can change its value from equation to equation. Its precise value can be determined based on the proofs. Furthermore, we make use of the notation $[n] := \{0, ..., n\}$ for $n \in \mathbb{N}$, and $[n]^k$ for a $k$-dimensional multi-index with range in $[n]$.

## 2 Universal lottery tickets

Before we can prove the existence of strong universal LTs, we have to formalize our notion of what makes a strong LT universal. First of all, a universal LT cannot exist in the same way as a strong LT, which is hidden in a randomly initialized deep neural network and is identified by pruning, i.e., setting a large amount of its parameters to zero while the rest keep their initial value. For a ticket to be universal and thus applicable to a variety of tasks, some of its parameters, if not all, need to be trained. So which parameters should that be? In deep transfer learning, it is common practice to only train the top layers (close to the output) of a large deep neural network. The bottom layers (close to the input) are reused and copied from a network that has been already trained successfully to perform a related task. This approach saves significant computational resources and often leads to improved training results. It is therefore reasonable to transfer it to LTs (Morcos et al., 2019).

Independently from LTs, we discuss conditions when this is a promising approach, i.e., when the bottom layers of the deep neural network represent multivariate (basis) functions, whose linear combination can represent a large class of multivariate functions. The independence of the functions is not required and could be replaced by dictionaries, but the independence aids the compression of the bottom layers and thus our objective to find sparse LTs. This view also provides an explanation of the empirically observed phenomenon that universal tickets achieve good performance across a number of tasks only at moderate sparsity levels and become more universal when trained on larger datasets (Morcos et al., 2019; Chen et al., 2020). Including a higher number of basis functions naturally reduces the sparsity of a LT but also makes it adaptable to richer function families.

### 2.1 How universal can a lottery ticket be?

**A trivial universal ticket.** A trivial solution of our problem would be to encode the identity function by the first layers, which would only require $2d$ or $d$ neurons per layer or even 1-2 neurons per convolutional layer. This would be an extremely sparse ticket, yet, pointless as the ticket does not reduce the hardness of our learning task. In contrast, it cannot leverage the full depth of the neural network and needs to rely on shallow function representations. How could we improve the learning task? The next idea that comes to our mind is to reduce the complexity of the function that has to be learned by the upper layers. For instance, we could restrict it to learn univariate functions. To explore this option, our best chance of success might be to utilize the following theorem.

The **Kolmogorov-Arnold representation theorem** states that every multivariate function can be written as the composition and linear combination of univariate functions. In particular, recent results based on Cantor sets $\mathcal{C}$ promise potential for efficient representations. Thm. 2 in (Schmidt-Hieber, 2021) shows the existence of only two univariate functions $g : \mathcal{C} \to \mathbb{R}$ and $\psi : [0,1] \to \mathcal{C}$ so that any continuous function $f : [0,1]^d \to \mathbb{R}$ can be written as $f(\boldsymbol{x}) = g\left(\sum_{i=1}^{d} 3^{1-i}\psi(x_i)\right)$. Furthermore, only $g$ depends on the function $f$, while $\psi$ is shared by all functions $f$ and is hence universal. Could $\sum_{i=1}^{d} 3^{1-i}\psi(x_i)$ be our universal LT? Unfortunately, it seems to be numerically infeasible to compute for higher input dimensions $d > 10$. In addition, the resulting representation of $f$ seems to be sensitive to approximation errors. On top of this, the outer function $g$ is relatively rough even though it inherits some smoothness properties of the function $f$ (cf. Schmidt-Hieber, 2021) and is difficult to learn. Thus, even restricting ourselves to learning an univariate function $g$ in the last layers does not adequately simplify our learning problem. To make meaningful progress in deriving a notion of universal LTs, we therefore need a stronger simplification.

### 2.2 Defining universality

To ensure that the knowledge of a LT substantially simplifies our learning task, we only allow the training of the last layer. A consequence of this requirement is that we have to limit the family of functions that we are able to learn, which means we have to make some concessions with regard to universality. We thus define a strongly universal LT always with respect to a family of functions.

We focus in the following on regression and assume that the last layer of a neural network has linear activation functions, which reduces our learning task to linear regression after we have established the LT. Classification problems could be treated in a similar way. Replacing the activation functions in the last layer by softmax activation functions would lead to the standard setting. In this case, we

would have to perform a multinomial logistic regression instead of a linear regression to train the last layer. We omit this case here to improve the clarity of our derivations.

**Definition 1** (Strong Universality). *Let $\mathcal{F}$ be a family of functions defined on $S \subset \mathbb{R}^d$ with $\mathcal{F} \ni g : S \to \mathbb{R}^n$. A function $b : \mathbb{R}^d \to \mathbb{R}^k$ is called strongly universal with respect to $\mathcal{F}$ up to error $\epsilon > 0$, if for every $f \in \mathcal{F}$ there exists a matrix $\boldsymbol{W} \in \mathbb{R}^{n \times k}$ and a vector $\boldsymbol{c} \in \mathbb{R}^n$ so that*

$$\sup_{x \in S} \|\boldsymbol{W}b(\boldsymbol{x}) + \boldsymbol{c} - f(\boldsymbol{x})\| \leq \epsilon. \tag{2}$$

Note that we have defined the universality property for any function, including a neural network. It also applies to general transfer learning problems, in which we train only the last layer. To qualify as a (strong) LT, we have to obtain $b$ by pruning a larger neural network.

**Definition 2** (Lottery ticket). *A neural network $f : \mathbb{R}^d \supset S \to \mathbb{R}^k$ is called a lottery ticket (LT) with respect to $f_0 : \mathbb{R}^d \supset S \to \mathbb{R}^k$ with parameters $\boldsymbol{\theta_0}$, if there exists a binary mask $\boldsymbol{B} \in \{0,1\}^{|\boldsymbol{\theta_0}|}$ so that $f_0(\boldsymbol{x}|\boldsymbol{B}\boldsymbol{\theta_0}) = f(\boldsymbol{x})$ for all $\boldsymbol{x} \in S$. We also write $f \subset f_0$.*

## 3 EXISTENCE OF UNIVERSAL LOTTERY TICKETS

Our Def. 1 of strong universality assumes that our target ticket $b$ has a finite amount of $k$ features, which is reasonable in practice but limits the complexity of the induced function family $\mathcal{F}$. However, universal function approximation regarding general continuous functions on $[0,1]^d$ can be achieved by neural networks only if they have arbitrary width (Pinkus, 1999; Cybenko, 1989; Kurt & Hornik, 1991) or arbitrary depth (Telgarsky, 2016; Yarotsky, 2017; Schmidt-Hieber, 2020). Feed forward networks of higher depth are usually more expressive (Yarotsky, 2018) and thus require less parameters than width constrained networks to approximate a continuous function $f$ with modulus of continuity $\omega_f$ up to maximal error $\epsilon$. Yarotsky (2018) has shown that the minimum number of required parameters is of order $O(\omega_f(O(\epsilon^{-d/2}))$ but has to assume that the depth of the network is almost linear in this number of parameters. Shallow networks in contrast need $O(\omega_f(O(\epsilon^{-d}))$ parameters. Note that the input dimension $d$ can be quite large in machine learning applications like image classification and the number of parameters depends on the Lipschitz constant of a function via $\omega_f$, which can be huge in general. In consequence, we need to narrow our focus regarding which function families we can hope to approximate with finite neural network architectures that have sparse representations and limit ourselves to the explicit construction of $k$ basis functions of a family that has the universal function approximation property.

We follow a similar strategy as most universal approximation results by explicitly constructing polynomials and Fourier basis functions. However, we propose a sparser, non-standard construction that, in contrast to the literature on feed forward neural networks, leverages convolutional layers to share parameters. Another advantage of our construction is that it is composed of linear multivariate and univariate functions, for which we can improve recent results on lottery ticket pruning. The existence of such sparse representations is remarkable because, in consequence, we would expect that most functions that occur in practice can be approximated by sparse neural network architectures and that these architectures are often universally transferable to other tasks.

**Polynomials** Sufficiently smooth functions, which often occur in practice, can be well approximated by a few monomials of low degree. At least locally this is possible, for instance, by a Taylor approximation. How can we approximate these monomials with a neural networks? In principle, we could improve on the parameter sparse construction by Yarotsky (2017); Schmidt-Hieber (2020) based on tooth functions by using convolutional layers that approximate univariate monomials in each component separately followed by feed forward fully-connected layers that multiply these pairwise. This, however, would require an unrealistically large depth $L = O(\log(\epsilon/k)\log(d))$ and also a considerable width of at least $n = O(kd)$ in many layers. Alternatively, we propose a constant depth solution as visualized in Fig. 1 (a). It has an $\epsilon$ dependent width that is maximally $n = O(d\sqrt{k/\epsilon})$ in just one layer and $n = O(\sqrt{kd/\epsilon})$ in another. It leverages the following observation. A multivariate monomial $b(\boldsymbol{x}) = \prod_{i=1}^{d} 0.5^{r_i}(1 + x_i)^{r_i}$, which is restricted to the domain $[0,1]^d$, can also be written as $b(\boldsymbol{x}) = \exp\left(\sum_i r_i \log(1 + x_i) - \log(2)\sum_i r_i\right)$. It is therefore a composition and linear combination of univariate functions $b(\boldsymbol{x}) = g\left(\sum_i r_i h(x_i)\right)$, where $g(x) = \exp(x)$ and $h(x) = \log(1 + x) - \log(2)$ as in Kolmogorov-Arnold form. Most importantly, every monomial

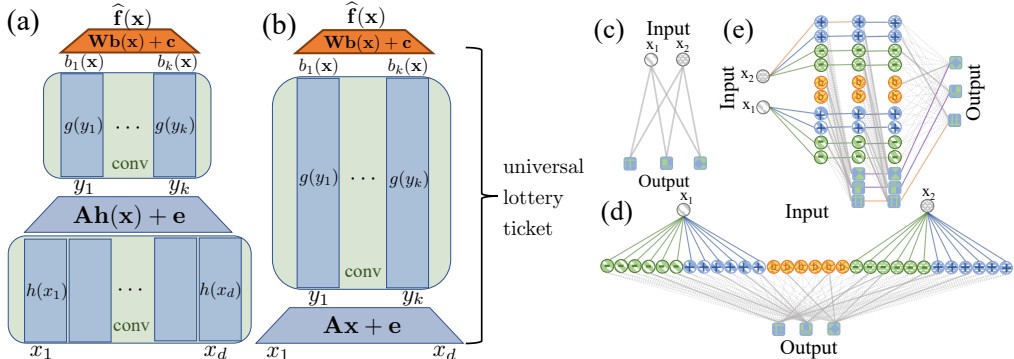

Figure 1: Left: Visualization of the proposed architecture for approximating function families. Potentially fully connected networks are colored blue, while green parts can be encoded as convolutional layers by applying the same function to all components. The last orange linear layer can be trained, while all other parameters are frozen to their initialization. (a) Polynomials: $h(x) = \log((1 + x)/2)$, $g(y) = \exp(y)$. (b) Fourier series: $g(x) = \sin(2\pi x)$. Right: Visualization of the lottery ticket construction for multivariate linear functions. (c) Visualization of an exemplary linear layer. (d) Approximation of the linear layer in (a) by a subset sum block. Blue nodes with label $+$ represent neurons in the intermediary layer that are pruned to neurons of the form $\phi(wx_{1/2})$ with $w > 0$, while green nodes correspond to neurons with $w < 0$, and orange nodes with label $b$ to bias neurons of the form $\phi(b)$. (e) The subset sum block in (d) is distributed across three layers. Two paths of bounded weight are highlighted in orange.

has the same structure. We can therefore construct a family of monomials efficiently by approximating each univariate function with a convolutional neural network and the linear combinations of the functions as potentially fully connected layers. Fig. 1 (a) visualizes the architecture and general construction idea. It applies to any composition of functions in form of the Kolmogorov-Arnold representation theorem, which in theory exists for every continuous multivariate function. This makes our construction more general than it may seem at first. In practice, however, the univariate functions still need to be amenable to efficient approximation by deep neural networks, which is the case for polynomials as we show in the next sections.

**Fourier series** Fourier analysis and discrete time Fourier transformations seek representations with respect to a Fourier basis $f(\boldsymbol{x}) = a_0 + \sum_{\boldsymbol{n} \in [N]^d} a_{\boldsymbol{n}} \sin(2\pi(\sum_{i=0}^{d} n_i x_i + c_n))$. Fig. 1 (b) shows how to construct the functions $\sin(2\pi(\sum_{i=0}^{d} n_i x_i + c_n))$ by (affine) linear combinations $\sum_{i=0}^{d} n_i x_i + c_n$ in the first layers close to the input followed by convolutional layers computing the univariate $\sin$, which has quite sparse representations if enough depth is available to exploit its symmetries. Again we use a composition of linear transformations and univariate functions, which share parameters by convolutional layers.

Even though the function families above can be represented efficiently as LTs, we should mention that a big advantage of neural networks is that the actual function family can be learned. This could also lead to a combination of different families, when this improves the ability of the lottery ticket to solve specific tasks efficiently. Accordingly, adding dependent functions to the outputs might also provide advantages. Our universal lottery ticket constructions allow for this as well. We simply focus our discussion on families of independent functions, as they usually induce higher sparsity.

### 3.1 EXISTENCE OF LOTTERY TICKETS LEVERAGING DEPTH

The targets that we propose as strongly universal functions are composed of linear and univariate neural networks. Some of our improvements with respect to the literature on LT existence leverage this fact. While Malach et al. (2020); Pensia et al. (2020); Orseau et al. (2020); Fischer & Burkholz (2021) provide a lower bound on the required width of the mother network $f_0$ so that a subnetwork could approximate our target neural network with depth $L_t$, $f_0$ would need to have exactly twice the depth $L_0 = 2L_t$, which would limit our universality statement to a specific architecture. To address

this issue, we allow for more flexible mother networks and utilize additional available depth. As Pensia et al. (2020), we approximate each target parameter by a subset sum block, but distribute this block across multiple network layers. This makes the original bound $n_0 = O(N \log (N/\epsilon))$ on the width of the mother network depth dependent.

This approach requires, among others, two additional innovations of practical consequence. The first one is our parameter initialization proposal. Most previous work neglects the biases and assumes that they are zero. Only Fischer & Burkholz (2021) can handle architectures with non-zero biases, which we need to represent our univariate functions of interest. They propose an initialization scheme that extends standard approaches like He (He et al., 2015) or Glorot (Glorot & Bengio, 2010) initialization to non-zero biases and supports the existence of LTs while keeping the large mother network $f_0$ trainable. We modify it to enable our second innovation, i.e., paths through the network that connect network subset sum blocks in different layers to the output.

**Definition 3** (Parameter initialization). *We assume that the parameters of a deep neural network are independently distributed as $w_{ij}^{(l)} \sim U\left([-\sigma_{w,l}, \sigma_{w,l}]\right)$ or $w_{ij}^{(l)} \sim \mathcal{N}\left(0, \sigma_{w,l}\right)$ for some $\sigma_{w,l} > 0$ and $b_i^{(l)} \sim U([-\prod_{k=1}^{l} \sigma_{w,k}/2, \prod_{k=1}^{l} \sigma_{w,k}/2])$ or $b_i^{(l)} \sim \mathcal{N}\left(0, \prod_{m=1}^{l} \sigma_{w,m}/2\right)$, respectively.*

Also dependencies between the weights as in (Burkholz & Dubatovka, 2019; Balduzzi et al., 2017) are supported by our proofs. The above initialization results in a rescaling of the output by $\lambda = \prod_{k=1}^{L} \sigma_{w,k}/2$ in comparison with an initialization of the weights by $w \sim U\left([-2, 2]\right)$ or $w \sim \mathcal{N}(0, 4)$ and biases by $b \sim U\left([-1, 1]\right)$ or $w \sim \mathcal{N}(0, 1)$. As pruning deletes a high percentage of parameters, we can expect that the output of the resulting network is also scaled roughly by this scaling factor $\lambda$ (see Thm. 2). However, the last linear layer that we concatenate to $f$ and assume to be trained can compensate for this. This rescaling means effectively that we can prune weight parameters from the interval $[-2, 2]$ in contrast to $[-1, 1]$ as in (Fischer & Burkholz, 2021; 2022). We can therefore find weight parameters $w_i$ that are bigger or smaller than 1 with high enough probability so that we can prune for paths of bounded weight $1 \leq \prod_{i=1}^{k} w_i \leq C$ through the network, as stated next.

**Lemma 1.** *Define $\alpha = 3/4$ and let $w_j \sim U[-2, 2]$ denote $k$ independently and identically (iid) uniformly distributed random variables with $j \in [k]$. Then $w_j$ is contained in an interval*

$$w_j \in \left[1/\left(\prod_{i=1}^{j-1} w_i\right), 1/\left(\alpha \prod_{i=1}^{j-1} w_i\right)\right]$$

*with probability at least $q = 1/16$. If this is fulfilled for each $w_j$, then*

$$1 \leq \left(\prod_{i=1}^{k} w_i\right) \leq 1/\alpha.$$

*The same holds true if each $w_j \sim \mathcal{N}(0, 4)$ is iid normally distributed instead.*

This defines the setting of our existence proofs as formalized in the next theorem.

**Theorem 2** (LT existence). *Assume that $\epsilon, \delta \in (0, 1)$, a target network $f \colon \mathcal{S} \subset \mathbb{R}^d \to \mathbb{R}^m$ with depth $L_t$ and architecture $\bar{n}_t$, and a mother network $f_0$ with depth $L_0 \geq 2$ and architecture $\bar{n}_0$ are given. Let $f_0$ be initialized according to Def. 3. Then, with probability at least $1 - \delta$, $f_0$ contains a sparse approximation $f_\epsilon \subset f_0$ so that each output component $i$ is approximated as $\max_{\boldsymbol{x} \in \mathcal{S}} |f_i(\boldsymbol{x}) - \lambda f_{\epsilon,i}(\boldsymbol{x})| \leq \epsilon$ with $\lambda = \prod_{l=1}^{L} (2\sigma_{w,l}^{-1})$ if $n_{l,0} \geq g(\bar{n}_t)$ for each $1 < l \leq L_0 - 1$.*

The required width $g(\bar{n}_t)$ needs to be specified to obtain complete results. We start with the construction of a single layer $\phi\left(W\boldsymbol{x} + \boldsymbol{b}\right)$, which we can also use to represent a linear layer by constructing its positive $\phi\left(W\boldsymbol{x} + \boldsymbol{b}\right)$ and negative part $\phi\left(-W\boldsymbol{x} - \boldsymbol{b}\right)$ separately. Note that in our polynomial architecture, all components of $W\boldsymbol{x} + \boldsymbol{b}$ are negative.

**Theorem 3** (Multivariate LTs (single layer)). *Assume the same set-up as in Thm. 2 and a target function $f(\boldsymbol{x}) = \phi\left(W\boldsymbol{x} + \boldsymbol{b}\right)$ with $M := \lceil \max_{i,j} \max(|w_{i,j}|, |b_i|) \rceil$, $N$ non-zero parameters, and $Q = (\sup_{\boldsymbol{x} \in \mathcal{S}} \|\boldsymbol{x}\|_1 + 1)$. A lottery ticket $f_\epsilon$ exists if*

$$n_{l,0} \geq C \frac{Md}{L_0 - 1} \log \left(M/\min \left\{\delta/(2(m+d)(L_0 - 1) + N + 1), \epsilon/Q\right\}\right)$$

*for each $1 < l \leq L_0 - 1$ whenever $L_0 > 2$ and $n_{1,0} \geq CMd \log \left( \frac{M}{\min\{\delta/(N+1), \epsilon/Q\}} \right)$ when $L_0 = 2$.*

*Proof idea.* The main idea to utilize subset sum approximations for network pruning has been introduced by Pensia et al. (2020). Each target layer $\phi(\boldsymbol{h}) = \phi(W\boldsymbol{x} + \boldsymbol{b})$ is represented by two layers in the pruned network $f_\epsilon$ as follows. Using the property of ReLUs that $x = \phi(x) - \phi(-x)$, we can write each pre-activation component as $h_i = \sum_{j=1}^m w_{ij}\phi(x_j) - w_{ij}\phi(-x_j) + \text{sign}(b_i)\phi(|b_i|)$. This suggests that intermediate neurons should be pruned into an univariate form $\phi(w_{lj}^{(1)} x_j)$ or $\phi(b_l^{(1)})$ if $b_l > 0$. Each actual parameter $w_{ij}$ or $b_i$ is, however, approximated by solving a subset sum approximation problem involving several intermediate neurons $l \in I_{ij}$ of the same type so that $w_{ij} \approx \sum_{l \in I_{ij}} w_{il}^{(2)} w_{lj}^{(1)}$. Fig. 1 (d) visualizes the approach. Node colors distinguish different neuron types $\phi(w_{lj}^{(1)} x_j)$ or a bias type $\phi(b_l^{(1)})$ for $b_l > 0$ within a subset sum block.

We split such a block into similarly sized parts and assign one to each available layer in the mother network as shown in Fig. 1 (e). The challenge in this construction is to collect the different results that are computed at different layers and to send them through the full network along a path. The product of weights $p = \prod_{k=1}^{L_0 - 2} w_k$ along this path is multiplied with the output. Since we can ensure that this factor is bounded reasonably by Lemma 1, we can compensate for it by solving a subset sum problem for $w_{ij} / \prod_{k=1}^{L_0 - 2} w_k$ instead of $w_{ij}$ without increasing the associated approximation error. Furthermore, we only need to prune $C \log(1/\delta')$ neurons in each layer to find such a path. A rigorous proof is presented in Appendix B.1. □

As every target neural network is composed of $L_t$ layers of this form, we have derived a bound $n_0 \geq O(n_t L_t / L_0 \log (n_t L_0 / (L_t \epsilon)))$ that scales as $1/L_0 \log(L_0)$ and highlights the benefits of additional depth in general LT pruning, which might be of independent interest. Yet, the depth of the mother network needs to be considerably larger than the one of the target. For univariate networks, we can again improve on this result in the following way. As explained in the appendix, every univariate neural network can be written as

$$f_N(x) = \sum_{i=0}^{N-1} a_i \phi(x - s_i) + a_N \tag{3}$$

with respect to $2N + 1$ parameters $a_i$ and $s_i$ for $0 \leq i \leq N - 1$, where the width $N$ determines its potential approximation accuracy of a continuous univariate function $f$ and enters our bound if we want to find a corresponding LT by pruning.

**Theorem 4** (Univariate LT). *Assume the same set-up as in Thm. 2 and that an univariate target network $f : \mathbb{S} \subset \mathbb{R} \to \mathbb{R}$ in form of Eq. (3) is given. Define $M := (1 + \max(\max_i |a_i|, \max_j |s_j|))$ and $Q = \max_{x \in S} |x|$. $f_0$ contains a LT $f_\epsilon \subset f_0$ if*

$$n_{l,0} \geq C \frac{\max\{M, N\}}{L_0 - 2} \log \left( \frac{M}{\min\{\delta/[L_0(N+2) - 1], \epsilon/(2(Q+M))\}} \right) \tag{4}$$

*for $L_0 \geq 3$. We require $n_{1,0} \geq C \frac{MQ}{\epsilon} \log \left( \frac{M}{\min\{\delta/(N+1), \epsilon/(2(Q+M))\}} \right)$ for $L_0 = 2$.*

Next, we can utilize our improved LT pruning results to prove the existence of universal LTs.

## 3.2 EXISTENCE OF POLYNOMIAL LOTTERY TICKETS

In the following, we discuss the existence of polynomial LTs in more detail. Corresponding results for Fourier basis functions are presented in the appendix. Our objective is to define a parameter sparse approximation of $k$ multivariate monomials $b(\boldsymbol{x}) = \prod_{i=1}^d 0.5^{r_i}(1 + x_i)^{r_i}$ as our target neural network (see Fig. 1 (a)) and then prove that it can be recovered by pruning a larger, but ideally not much larger, randomly initialized mother network. Thus, in contrast to previous results on LT existence, we have to take not only the pruning error but also the approximation error into account.

A multivariate linear function can be represented exactly by a ReLU neural network, but we have to approximate the known functions $h(x) = \log((1 + x)/2)$ and $g(y) = \exp(y)$ in a parameter sparse

way. Thm. 2 Yarotsky (2017) states that univariate Lipschitz continuous function can be approximated by a depth-6 feed forward neural network with no more than $c/(\epsilon \log(1/\epsilon))$ parameters. By specializing our derivation to our known functions, in contrast, we obtain a better scaling $c/\sqrt{\epsilon}$ with depth $L = 2$, which leads to our main theorem.

**Theorem 5.** *Let $\mathcal{B} \subset \left\{ \prod_{i=1}^{d} 0.5^{r_i}(1 + x_i)^{r_i} | r_i \leq q, \sum r_i \leq t, x_i \in [0, 1] \right\}$ be a subset of the polynomial basis functions with bounded maximal degree $q$ of cardinality $k = |\mathcal{B}|$. Let $\mathcal{F}(\mathcal{B})$ be the family of bounded (affine) linear combinations of these basis functions. For $\epsilon, \delta \in (0, 1)$, with probability $1 - \delta$ up to error $\epsilon$, $f_0$ contains a strongly universal lottery ticket $f_\epsilon \subset f_0$ with respect to $\mathcal{F}$ if $f$ has $L_{log} \geq 2$ convolutional layers, followed by $L_{multi} \geq 2$ fully connected layers, and $L_{exp} \geq 2$ convolutional layers with channel size or width chosen as in Thms. 4 and 3 with the following set of parameters.*

*Logarithm: $\delta_{log} = 1 - (1 - \delta)^{1/3}$, $\epsilon_{log} = \frac{\epsilon}{6tkm}$, $N_{log} = 1 + \left\lceil \sqrt{\frac{tkm}{\epsilon}} \right\rceil$, $M_{log} = 2$, $Q_{log} = 1$.*

*Multivariate: $\delta_{multi} = 1 - (1 - \delta)^{1/3}$, $\epsilon_{multi} = \frac{\epsilon}{6km}$, $M_{multi} = q$, $Q_{multi} = d$.*

*Exponential: $\delta_{exp} = 1 - (1 - \delta)^{1/3}$, $\epsilon_{exp} = \frac{\epsilon}{6km}$, $N_{exp} = 1 + \left\lceil t\sqrt{\frac{km}{4\epsilon}} \right\rceil$, $M_{exp} = 2$, $Q_{exp} = t \log 2$.*

As we show next in experiments, all our innovations provide us with practical insights into conditions under which pruning for universal lottery tickets is feasible.

## 4 EXPERIMENTS

To showcase the practical relevance of our main theorems, we conduct two types of experiments on a machine with Intel(R) Core(TM) i9-10850K CPU @ 3.60GHz processor and GPU NVIDIA GeForce RTX 3080 Ti. First, we show that the derived bounds on the width of the mother networks are realistic and realizable by explicitly pruning for our proposed universal lottery tickets. Second, we prune mother network architectures from our theorems for strong winning tickets with the edge-popup algorithm (Ramanujan et al., 2020) to transfer our insights to a different domain.

In the first type of experiment, we explicitly construct universal lottery tickets by following the approach outlined in our proofs. The construction involves solving multiple subset sum approximation problems, which is generally NP-hard. Instead, we obtain an good approximate solution by taking the sum over the best 5-or-less-element subset out of a random ground set. Usually, ground sets consisting of 20 to 30-elements are sufficient to obtain good approximation results. The mother network has uniformly distributed parameters according to Def. 3.

**Polynomial function family:** We construct a polynomial function family consisting of $k$ features of the form $(1 + x_i)^b$ for any $b \in [0, 4]$ assuming $d$-dimensional inputs. If the mother network has a maximum channel size of $500$, we obtain a maximum approximation error of $0.001$, which is negligible. Concretely, assuming convolutional (or fully connected) layers of sizes $[d, 200, 10, 200, 10, k * 30, k, 500, 40, 500, 1]$ (or higher) is sufficient. This further assumes that we use $N_{\log} = 10$ and $N_{\exp} = 20$ intermediary neurons in the approximation of the respective univariate functions. After pruning, $0.022$ of the original parameters remain, which is sparser than the tickets that most pruning algorithms can identify in practice. Note that we can always achieve an even better sparsity relative to the mother network by increasing the width of the mother network.

**Fourier basis**: The most restrictive part is the approximation of $f(x) = \sin(2\pi x)$ on $[0, 1]$, which can be easily obtained by pruning a mother network of depth $4$ with architecture $[1, 250, 21, 250, 1]$ with $N_{\sin} = 21$ intermediary neurons in the univariate representation. Pruning such a random network achieves an approximation error of $0.01$ and keeps only $0.038$ of the original parameters. Note that we can always achieve a better sparsity by increasing the width of the mother network. For instance, pruning a network with architecture $[1, 250, 250, 250, 1]$ would result in $0.0035$.

In the second type of experiments, we train our mother networks with edge-popup (Ramanujan et al., 2020) on MNIST (LeCun & Cortes, 2010) for 100 epochs based on SGD with momentum 0.9, weight decay 0.0001, batch size 128, and target sparsity 0.5. Parameters are initialized from a normal distribution according to Def. 3. As the original algorithm is constrained to zero biases, we had to extend it to pruning non-zero biases as well. It finds subnetworks of the randomly initialized mother ticket architecture achieving an average accuracy with 0.95 confidence region of $92.4 \pm 0.5$

% (polynomial architecture) and $97.9 \pm 0.1$ % (Fourier architecture) over 5 independent runs. Note that pruning the polynomial architecture is not always successful on MNIST but such runs can be detected early in training. This implies that our insights can transfer to a domain different from polynomial or Fourier regression and attests the architectures some degree of universality. Note that even though edge-popup is free to learn tickets that are different from the proposed universal function families, the architecture is still constrained to compositions of univariate and linear multivariate functions.

In addition, these experiments highlight that our parameter initialization (with non-zero biases) induces trainable architectures, as the success of edge-popup relies on meaningful gradients.

## 5 DISCUSSION

We have derived a formal notion of strong universal lottery tickets with respect to a family of functions, for which the knowledge of a universal lottery ticket reduces the learning task to linear regression. As we have proven, these universal lottery tickets exist in a strong sense, that is, with high probability they can be identified as subset of a large randomly initialized neural network. Thus, once a ticket is identified, no further training is required except for the linear regression. We have shown this with an explicit construction of deep ReLU networks that represent two major function classes, multivariate polynomials and trigonometric functions, which have the universal approximation property.

These classes consist of basis functions which can be represented by compositions of univariate functions and multivariate linear functions, which are amenable to sparse approximations by deep ReLU networks. As we highlight, the use of convolutional network layers can significantly improve the sparsity of these representations. Up to our knowledge, we have presented the first proof of the existence of universal lottery tickets and of lottery tickets in general in convolutional neural networks. Most remarkable is the fact that these common function classes with the universal approximation property have sparse neural network representations. Furthermore, they can be found by pruning for (universal) lottery tickets. In consequence, we should expect that many tasks can be solved with the help of sparse neural network architectures and these architectures transfer potentially to different domains.

Our theoretical insights provide some practical guidance with respect to the setting in which sparse universal tickets could be found. We have shown how parameters can be initialized effectively and discussed what kind of architectures promote the existence of universal lottery tickets. In comparison with the theoretical literature on strong lottery tickets, we have relaxed the requirements on the width of the large randomly initialized mother network and made its depth variable. Some of our novel proof ideas might therefore be of independent interest for lottery ticket pruning. Interestingly, in contrast to the standard construction of lottery tickets, we derived a proof that does not start from representing a function as neural network but as linear combination with respect to a basis. In future, we might be able to use similar insights to deepen our understanding of what kind of function classes are amenable to training by pruning in search of lottery tickets.

As a word of caution, we would like to mention that a strongly universal ticket (whose existence we have proven) is not necessarily also a 'weakly' universal ticket (which is more commonly identified in practice). The reason is that in case that we train a ticket that represents the right basis functions together with the last layer (i.e. the linear regression weights), also the parameters of the ticket might change and move away from the suitable values that they had initially. In several common initialization settings, however, the parameters of the bottom layers (that correspond to the ticket) change only slowly during training, if at all. Therefore, a strong ticket will often be also a weak ticket from a practical point of view.

Furthermore, it is important to note that the universal lottery tickets that we have constructed here are not necessarily the tickets that are identified by pruning neural networks in common applications related to imaging or natural language processing. It would be interesting to see in future how much redundancy these tickets encode and whether they could be further compressed into a basis.

ETHICS AND REPRODUCIBILITY STATEMENT

We do not have ethical concerns about our work that go beyond the general risks associated with deep learning that are, for instance, related to potentially harmful applications, a lack of fairness due to biases in data, or vulnerabilities of deployed models to adversarial examples. The advantage of universal lottery tickets, whose existence we have proven here, is that we might be able to understand and address some of this limitations for specific architectures that can than be reused in different contexts. Code for the experiments is publicly available in the Github repository UniversalLT, which can be accessed with the following url `https://github.com/RelationalML/UniversalLT`.

ACKNOWLEDGMENTS

NL and RM were supported by the National Institutes of Health grant P42ES030990. AG was supported by a Swiss National Science Foundation Early Postdoc.Mobility fellowship.

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

## A   RESULTS IN SUPPORT OF EXISTENCE THEOREMS

In the following sections, we represent the proofs of our theorems and discuss different representation options of lottery tickets. In particular, we highlight the benefits of additional depth for reducing the maximum required width of the mother network $f_0$. We frequently utilize various forms of solutions to the subset sum problem and present novel results with regard to this general problem.

### A.1   INITIALIZATION

We derive our proofs by assuming that the parameters of the randomly initialized mother network $f_0$ are distributed in the following way. In case of uniform parameter initialization, we assume that the weights $w$ and biases $b$ follow the distributions $w \sim U[-2, 2]$ and biases $b \sim U[-1, 1]$. In case of normal parameter initialization, we have $w \sim \mathcal{N}(0, 4)$ and biases $b \sim \mathcal{N}(0, 1)$. The following lemma explains, why we can follow this approach when the parameters are initialized according to Definition 3 and the output of a neural network is corrected by a scaling factor $\lambda = \prod_{l=1}^{L}(2\sigma_{w,l}^{-1})$.

**Lemma 6** (Output scaling). *Let $h(\boldsymbol{\theta_0}, \boldsymbol{\sigma})$ denote a transformation of the parameters $\boldsymbol{\theta_0}$ of the deep neural network $f_0$, where each weight is multiplied by a scalar $\sigma_l$, i.e., $h_{ij}^{(l)}(w_{0,ij}^{(l)}) = \sigma_l w_{0,ij}^{(l)}$, and each bias is transformed to $h_i^{(l)}(b_{0,i}^{(l)}) = \prod_{m=1}^{l} \sigma_m b_{0,i}^{(l)}$. Then, we have $f(x \mid h(\boldsymbol{\theta_0}, \boldsymbol{\sigma})) = \prod_{l=1}^{L} \sigma_l f(x \mid \boldsymbol{\theta_0})$.*

For completeness, we present the proof but note that it has been also derived by (Fischer & Burkholz, 2021).

*Proof.* Let the activation function $\phi$ of a neuron either be a ReLU $\phi(x) = \max(x, 0)$ or the identity $\phi(x) = x$. A neuron $x_i^{(l)}$ in the original network becomes $g\left(x_i^{(l)}\right)$ after parameter transformation. We prove the statement by induction over the depth $L$ of a deep neural network.

First, assume that $L = 1$ so that we have $x_i^{(1)} = \phi\left(\sum_j w_{ij}^{(1)} x_j + b_i^{(1)}\right)$ After transformation by $w_{ij}^{(1)} \mapsto \sigma_1 w_{ij}^{(1)}$ and $b_i^{(1)} \mapsto \sigma_1 b_i^{(1)}$, we receive $g\left(x_i^{(1)}\right) = \phi\left(\sum_j w_{ij}^{(1)} \sigma_1 x_j + \sigma_1 b_i^{(1)}\right) = \sigma_1 x_i^{(1)}$ because of the homogeneity of $\phi(\cdot)$. This proves our claim for $L = 1$.

Next, our induction hypothesis is that $g\left(x_i^{(L-1)}\right) = \prod_{m=1}^{L-1} \sigma_m x_i^{(L-1)}$. It follows that

$$g\left(x_i^{(L)}\right) = \phi\left(\sum_j w_{ij}^{(L)} \sigma_L g\left(x_j^{(L-1)}\right) + b_i^{(L)} \prod_{m=1}^{L} \sigma_m\right) \quad \text{(def. of transformation)} \quad (5)$$

$$= \phi\left(\sum_j w_{ij}^{(L)} \sigma_L \prod_{m=1}^{L-1} \sigma_m x_j^{(L-1)} + b_i^{(L)} \prod_{m=1}^{L} \sigma_m\right) \quad \text{(induction hypothesis)} \quad (6)$$

$$= \prod_{m=1}^{L} \sigma_m x_i^{(L)} \quad \text{(homogeneity of } \phi), \quad (7)$$

which was to be shown.                                                                 $\square$

## A.2 SUBSET SUM PROBLEM

Our proofs frequently rely on solving the subset sum problem (Lueker, 1998), which was first utilized in existence proofs of strong lottery tickets without biases (Pensia et al., 2020) and then transferred to proofs of strong tickets with biases (Fischer & Burkholz, 2021; 2022). It applies to general iid random variables that contain a uniform distribution, as defined next.

**Definition 4.** *A random variable $X$ contains a uniform distribution if there exist constants $\alpha \in (0,1]$, $c, h > 0$ and a random variable $G_1$ so that $X \sim \alpha U[c-h, c+h] + (1-\alpha)G_1$.*

Such random variables can be used to approximate any $z \in [-m, m]$ in a bounded interval, as our next corollary states.

**Corollary 7** (Subset sum approximation ). *Let $X_1, ..., X_n$ be independent bounded random variables with $|X_k| \leq B$. Assume that each $X_k \sim X$ contains a uniform distribution with $c = 0$ (see Definition 4). Let $\epsilon, \delta \in (0,1)$ and $m \in \mathbb{R}$ with $m \geq 1$ be given. Then for any $z \in [-m, m]$ there exists a subset $S \subset [n]$ so that with probability at least $1 - \delta$ we have $|z - \sum_{k \in S} X_k| \leq \epsilon$ if*

$$n \geq C \frac{\max\left\{1, \frac{m}{h}\right\}}{\min_k\{\alpha_k\}} \log\left(\frac{B}{\min\left(\frac{\delta}{\max\{1, m/h\}}, \frac{\epsilon}{\max\{m, h\}}\right)}\right).$$

*Proof.* For $m = 1$ the proof is a subset of the proof of Corollary 3.3. by Lueker (1998). To extend these results to $m > 1$, let us fix a $\delta'$ and $\epsilon'$ that we will later choose depending on $\epsilon$, $\delta$, and $m$. We know that approximating any $z' \in [-1, 1]$ is feasible with probability $1 - \delta'$ up to error $\epsilon'$ based on random variables $X_k/h$ as long as we have at least

$$n_1 \geq C \frac{h}{\alpha} \log\left(\frac{B}{\min(\delta', \epsilon')}\right). \tag{8}$$

random variables.

Next we distinguish two different cases. 1) Let us start with $h/m \leq 1$. To approximate any $z \in [-m, m]$, we approximate $z' = zh/m$ by solving $m' = \lceil m/h \rceil \geq 1$ separate subset sum approximation problems. Note that the variables $X_k/h$ contain a uniform distribution $U[-1, 1]$ so that we can approximate $z/m \in [-1, 1]$ by $|z/m - \sum_{k \in S} X_k/h| \leq \epsilon'$. It follows that $|zh/m - \sum_{k \in S} X_k| \leq \epsilon' h$. We use this approximation result $m' = \lceil m/h \rceil$ times. Accordingly, we draw in total $n = \lceil m/h \rceil n_1$ random variables, assign each to one of $m' = \lceil m/h \rceil$ independent batches $A_k = (X_{(k-1)m'+1}, ..., X_{km'})$, and use each batch $A_k$ to approximate $zh/m$ up to error $\epsilon' h$. This approximation is successful for all batches with probability $(1-\delta')^{m'}$ and identifies index sets $S_k$ so that

$$\left|z - \sum_{i \in \cup_k S_k} X_i\right| \leq \sum_{k=1}^{m'} \left|zh/m - \sum_{i \in S_k} X_i\right| \leq m'h\epsilon'.$$

Thus, if we choose $\delta'$ and $\epsilon'$ so that $(1-\delta')^{m'} \geq (1-\delta)$, we obtain the desired approximation guarantees with $n = m'n_1$. This is fulfilled for $\delta' = \delta/m' \approx \delta h/m$, and $\epsilon' = \epsilon/(hm') \approx \epsilon/m$.

2) The second case assumes $h/m > 1$. Let us define $\tilde{z} := zm/h$ with $m/h < 1$, for which we have $|\tilde{z}/m| < |z|/m \leq 1$. Thus, also $z/h \in [-1, 1]$, as $z/h = \tilde{z}/m$. It follows that we can approximate $z/h$ with a subset $S$ of $n_1$ random variables $X_k/h$ so that $|z/h - \sum_{k \in S} X_k/h| \leq \epsilon'$. Hence, we have

$$\left|z - \sum_{k \in S} X_k\right| \leq h\epsilon'.$$

The choice $\epsilon' = \epsilon/h$ meets our approximation objective. $\qquad \square$

Next, we explicitly state two special cases that we use frequently in our initialization set-up.

**Lemma 8** (Approximation by products of uniform random variables). *Let $X_1, ..., X_n$ be iid random variables with a distribution $X_1 \sim V_2 V_1$, where $V_1 \sim U[0,1]$ and $V_2 \sim U[-2,2]$ are independently*

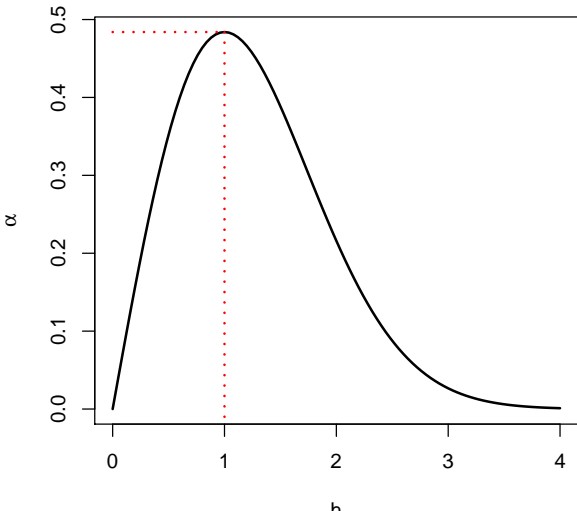

Figure 2: Each standard normal distribution $Z \sim \mathcal{N}(0,1)$ contains a uniform distribution $U[-h, h]$ with the shown $\alpha$, where $\alpha = \phi(h/\sigma)2h/\sigma$. We use the highlighted value $h = 1$ in our proofs.

*distributed. Let $\epsilon, \delta \in (0,1)$ and $m > 0$ be given. Then for any $z \in [-m, m]$ there exists a subset $S \subset [n]$ so that with probability at least $1 - \delta$ we have $|z - \sum_{i \in S} X_i| \leq \epsilon$ if*

$$n \geq Cm \log \left( \frac{m}{\min(\delta, \epsilon)} \right). \tag{9}$$

*Proof.* Our main proof strategy is the application of Corollary 7. Since $X_1$ is obviously bounded by $B = 2$, we only have to show that $X_1$ contains a uniform distribution. For $V_2 \sim U[-1, 1]$ this has been shown already by Pensia et al. (2020) with Corollary 1 and $\alpha = \log(2)/2$, $h = 1/2$, and $c = 0$. The same arguments apply to our case with $\alpha = \log(4)/4$, $h = 1$, and $c = 0$, which we integrate into the generic constant $C$. $\qquad \square$

Alternatively, we might want to consider initialization schemes with normally distributed parameters. The next lemma confirms that each normal distribution contains a uniform distribution.

**Lemma 9.** *A normally distributed random variable $Z \sim \mathcal{N}(0, \sigma^2)$ contains a uniform distribution $U[-\sigma, \sigma]$ with $\alpha = 0.4$.*

*Proof.* From Definition 4 it can be showed that a random variable $Z$ contains a uniform distribution $U[-h, h]$ if the probability that $Z \in [-h, h]$ is at least as big as if would flip a biased coin that turns up heads with probability $\alpha > 0$. In case of this event, we would further draw an element from $[-h, h]$ from a uniform distribution $U[-h, h]$. In other words, it suffices to show that there exists $\alpha > 0$ so that the probability density $p_Z(x)$ fulfills $p_Z(x) \geq \alpha/(2h)$ for all $x \in [-h, h]$.

Since $Z \sim \mathcal{N}(0, \sigma^2)$, its density is given by $\phi(x/\sigma)/\sigma$, where $\phi(x)$ denotes the density of the standard normal distribution. Since $\phi(x)$ is monotonously decreasing in $x$ for $x > 0$, we need to fulfill $\alpha \leq \phi(h/\sigma)2h/\sigma$. For $h = \sigma$, this is fulfilled for $\alpha = 0.4 < 2 * \phi(1)$. Figure 2 shows the different choices of $\alpha$ for varying $h$ and $\sigma = 1$, which confirms that $h = \sigma$ in general leads to relatively high values of $\alpha$. $\qquad \square$

We can use this proof to show that products of normal distributions are also suitable for approximation with the help of the subset sum problem.

**Lemma 10** (Approximation by products of normal random variables). *Let $X_1, ..., X_n$ be iid random variables with a distribution $X_1 \sim V_2V_1$, where $V_1 \sim (N \mid N > 0)$ with $N \sim (\mathcal{N}(0, \sigma_1^2))$ and $V_2 \sim \mathcal{N}(0, 4)$ are independently distributed with $\sigma_1 = 1$ or $\sigma_1 = 2$. Let $\epsilon, \delta \in (0, 1)$ and $m > 0$ be given. Then for any $z \in [-m, m]$ there exists a subset $S \subset [n]$ so that with probability at least $1 - \delta$ we have $|z - \sum_{i \in S} X_i| \leq \epsilon$ if*

$$n \geq Cm \log \left( \frac{m}{\min(\delta, \epsilon)} \right). \tag{10}$$

*Proof.* To apply Corollary 7, we have to show that the product $V_1V_2$, contains a uniform distribution. In addition, we have to solve the technical problem that normal distributions are not bounded random variables. However, for a given bound $B$, we can simply ignore all variables that exceed that bound $|X_i| > B$ and just make sure that enough bounded variables are left with high probability so that we can solve our approximation problem. In fact, let us only use variables with $1/2 \leq V_1 \leq 1$ and $|V_2| \leq B$ and assume that we need at least $n_p$ of the variables $X_i$ that fulfill these criteria with probability $(1 - \delta/2)$. With the help of a union bound, we can see that we need at least $n \geq C(n_p + \log(2/\delta))$ variables to ensure this. Note that $C$ depends on the bound $B$ and decreases for increasing $B$, while the width requirement in Corollary 7 increases in $B$ as $\log(B)$. One could trade-off both with the right choice of $B$ but this is not relevant conceptually in our proofs.

Given the variable $V_1$, the random variable $V_2V_1$ is distributed as $V_2V_1 \mid V_1 \sim \mathcal{N}(0, 4V_1^2)$, which contains $U[-2V_1, 2V_1]$ with $\alpha = 0.4$ according to Lemma 9. Note that $U[-2V_1, 2V_1]$ contains $U[-1, 1]$ for all $1/2 \leq V_1 \leq 1$ with $\alpha = 1/(2V_1)$ so that $V_2V_1 \mid V_1$ contains $U[-1, 1]$ with $\alpha = 0.4/(2V_1) \geq 0.2$ Thus, $V_2V_1$ also contains $U[-1, 1]$ with $\alpha = 0.2$.

Applying Corollary 7 to bound $n_p$ thus concludes our proof. □

Next, we will prove an even stronger result on approximation by solving a subset sum problem that allows us to mix random variables with different distributions.

**Corollary 11** (Extended approximation with subset sum). *Let $X_1, ..., X_{n_x}$ be independent bounded random variables with $|X_i| \leq B$. Assume that each $X_i \sim X$ contains a uniform distribution $U[-1, 1]$ with potentially different $\alpha_i > 0$ (see Definition 4). Let $\epsilon, \delta \in (0, 1)$ and $m \in \mathbb{N}$ with $m \geq 1$ be given. Then for any $z \in [-m, m]$ there exists a subset $S \subset [n]$ so that with probability at least $1 - \delta$ we have $|z - \sum_{i \in S} X_i| \leq \epsilon$ if*

$$n \geq C \frac{hm}{\min_i\{\alpha_i\}} \log \left( \frac{Bm}{\min(\delta, \epsilon)} \right). \tag{11}$$

*Proof.* Following the same arguments as Lueker (1998) in his proof of Corollary 3.3, we only need to ensure that we can obtain $k \geq C \log(1/\epsilon)$ samples that are drawn from a uniform distribution $U[-1, 1]$ to approximate a $z \in [-1, 1]$ up to precision $\epsilon$ with high probability. The probability that an individual sample $X_i$ falls into this category is $\alpha_i$ and is at least $\alpha' = \min_i\{\alpha_i\}$ for every sample. With this $\alpha'$, we can follow exactly the same steps as Lueker (1998) and arrive at the stated result. □

The following lemma is also stated in the main manuscript as Lemma 1.

**Lemma 12.** *Define $\alpha = 3/4$ and let $w_j \sim U[-2, 2]$ denote $k$ iid uniformly distributed random variables with $j \in [k]$. Then $w_j$ is contained in an interval*

$$w_j \in \left[ 1/ \left( \prod_{i=1}^{j-1} w_i \right), 1/ \left( \alpha \prod_{i=1}^{j-1} w_i \right) \right]$$

*with probability at least $q = 1/16$. If this is fulfilled for each $w_j$, then*

$$1 \leq \left( \prod_{i=1}^{k} w_i \right) \leq 1/\alpha.$$

*The same holds true if each $w_j \sim \mathcal{N}(0, 4)$ is iid normally distributed instead.*

*Proof.* We prove the statement by induction. First, let $k = 1$ and note that $1 \leq w_1 \leq 1/\alpha$ with probability $(4/3 - 1)/4 = 1/12 \geq 1/16$ for uniformly distributed $w_1$. For normally distributed $w_1$, $1 \leq w_1 \leq 1/\alpha$ is fulfilled with probability $\Phi(1/(2\alpha))) - \Phi(1/2) \geq (1/\alpha - 1)/4 = 1/12 \geq 1/16$.

In the induction step from $k$ to $k + 1$, we assume that

$$w_j \in \left[ 1/\left( \prod_{i=1}^{j-1} w_i \right), 1/\left( \alpha \prod_{i=1}^{j-1} w_i \right) \right]$$

with probability $q$ for all $j \leq k$, which implies that

$$1 \leq \left( \prod_{i=1}^{k} w_i \right) \leq 1/\alpha.$$

We have to show that also $1 \leq w_{k+1} \left( \prod_{i=1}^{k} w_i \right) \leq 1/\alpha$ and thus

$$w_{k+1} \in \left[ 1/\left( \prod_{i=1}^{k} w_i \right), 1/\left( \alpha \prod_{i=1}^{k} w_i \right) \right].$$

For uniformly distributed $w_{k+1}$, this is fulfilled with probability $(1/\alpha - 1)/\left( 4 \prod_{i=1}^{k} w_i \right) \geq (1/\alpha - 1)\alpha/4 = 1/16 = q$, where we used the induction hypothesis. For normally distributed $w_{k+1}$, this is also fulfilled with probability $(1/\alpha - 1)\alpha/4 = 1/16 \geq q$. $\qquad\square$

## B  EXISTENCE OF LINEAR AND UNIVARIATE LOTTERY TICKETS

At least for specific function classes, we can significantly relax the width and depth requirements. The family of functions that we are particularly interested in are compositions of univariate functions and multivariate linear functions. This also allows us to leverage the parameter sharing offered by convolutional layers to gain further improvements. We first focus on fully-connected mother networks $f_0$, as we can also utilize them along the channel dimension in convolutional layers as well.

### B.1  MULTIVARIATE LINEAR LOTTERY TICKETS

Our first existence results we present for affine linear multivariate functions $f(\boldsymbol{x}) = W\boldsymbol{x} + \boldsymbol{b}$. As every layer of a neural network layer is essentially a composition of this function and an activation function $\phi(f(\boldsymbol{x}))$, the main ideas that we present here could be immediately transferred to pruning for general target networks.

**Statement.** *Assume that $\epsilon, \delta \in (0, 1)$ and a target function $f : \mathbb{R}^d \supset S \to \mathbb{R}^m$ with $f(\boldsymbol{x}) = W\boldsymbol{x} + \boldsymbol{b}$ with $M := \lceil \max_{i,j} \max(|w_{i,j}|, |b_i|) \rceil$, $N$ non-zero parameters, and $Q = (\sup_{\boldsymbol{x} \in S} \|\boldsymbol{x}\|_1 + 1)$ are given. Let each weight and bias of $f_0$ with depth $L \geq 2$ and architecture $\bar{n}_0$ be initialized according to Def. 3. Then, with probability at least $1 - \delta$, $f_0$ contains a sparse approximation $f_\epsilon \subset f_0$ so that each output component $i$ is approximated as $\max_{\boldsymbol{x} \in S} |f_i(\boldsymbol{x}) - \lambda f_{\epsilon,i}(\boldsymbol{x})| \leq \epsilon$ if*

$$n_{l,0} \geq C \frac{Md}{L-1} \log \left( M / \min \{\delta/(2(m+d)(L-1) + N + 1), \epsilon/Q\} \right)$$

*for $L > 2$ and*

$$n_{1,0} \geq CMd \log \left( \frac{M}{\min \{\delta/(N+1), \epsilon/Q\}} \right)$$

*for $L = 2$ and $\lambda = \prod_{l=1}^{L}(2\sigma_{w,l}^{-1})$.*

*Proof.* First, we analyze the amount of error $\epsilon_l$ that we can make with each parameter and still meet our approximation objective.

**Error propagation**:

Let us assume we approximate a target function component $f_i(\boldsymbol{x}) = \boldsymbol{w}^T\boldsymbol{x} + b_i$ with another multivariate affine linear function $g_\epsilon(\boldsymbol{x}) = \boldsymbol{a}^T\boldsymbol{x} + c$ with close parameters so that $|a_i - w_i| \leq \epsilon_l$ and $|b_i - c| \leq \epsilon_l$. Then, it follows that $\sup_{\boldsymbol{x} \in \mathcal{S}} |f_i(\boldsymbol{x}) - g_{\epsilon,i}(\boldsymbol{x})| \leq \sup_{\boldsymbol{x} \in \mathcal{S}} \sum_{j=1}^d |w_j - a_j||x_j| + |b_i - c| \leq \epsilon_l \left(\sup_{\boldsymbol{x} \in \mathcal{S}} \|\boldsymbol{x}\|_1 + 1\right) = Q\epsilon_l \leq \epsilon$, if $\epsilon_l \leq \epsilon/Q$.

**Lottery ticket construction**:

Next, we have to construct a lottery ticket that computes such a function $g_\epsilon(\boldsymbol{x})$ for each output component. Our construction strategy depends on the available depth $L$. We start with the case $L = 2$.

**Case $L = 2$**:

As Pensia et al. (2020), we utilize solutions to the subset sum problem. In addition, however, we have to deal with the fact that our parameters are not necessarily bounded by $1$ and that we have non-zero biases. While Layer 0 and Layer 2 of both $f_0$ and our ticket $f_\epsilon$ are fixed to the input and output neurons respectively, we can only prune Layer 1 of $f_0$, which has width $n_{0,1}$. How could we represent our target $f$ in this setting? We can write each output component of our target as

$$f_i(\boldsymbol{x}) = \sum_{j=1}^d w_j x_j + b_i = \sum_{j=1}^d w_j \phi(x_j) + \sum_{j=1}^d (-w_j)\phi(-x_j) + b_i\phi(1),$$

utilizing the identity $x = \phi(x) - \phi(-x)$ for ReLUs. This shows that we can represent $f$ as 2-layer neural network consisting of $2d + 1$ neurons in Layer 1, i.e., $\phi(x_j)$, $\phi(-x_j)$, and $\phi(1)$.

Let us thus prune $n_p$ out of the $n_{0,1}$ neurons in $f_0$ to neurons of the form $u_j\phi(v_j x_j)$ with $v_j > 0$ to help represent $w_j\phi(x_j)$ and prune $n_p$ of the neurons in $f_0$ to neurons of the form $u_j\phi(v_j x_j)$ with $v_j < 0$ to help represent $(-w_j)\phi(-x_j)$ and prune $n_p$ neurons to $u_j\phi(v_j)$ with $v_j > 0$ to represent $b_i\phi(1)$.

How large must $n_{0,1}$ be so that this pruning is feasible with probability at least $1 - \delta'$ for a given $n_p$ and $\delta' > 0$? (Note that this was assumed to be possible in (Pensia et al., 2020) with probability one, which is not entirely correct.) To use the union bound to answer this question, let us first estimate the probability that we cannot find $n_p$ neurons of each type. If we could not find those then at least $n_{0,1} - n_p + 1$ neurons in $f_0$ must be useless to prevent pruning of a neuron of a specific type. A neuron is useless for a specific type with probability $0.5$, since $v_j > 0$ (or $v_j > 0$) with probability $0.5$. We only have $(d + 1)$ possible cases of failed pruning, because pruning can only fail for one of the types $w_j\phi(x_j)$ or $(-w_j)\phi(-x_j)$ but not for both, since each neuron in Layer 1 of $f_0$ falls in one of the two categories. It follows that with the help of a union bound, we can thus ensure that we can find enough neurons of each type with probability at least $1 - \delta'$ if $1 - \delta' = 1 - (d+1)0.5^{n_{0,1}-n_p+1}$ and thus

$$n_{0,1} \geq \max\left(n_p + \log(2)\log\left((d+1)/\delta'\right), (2d+1)n_p\right). \tag{12}$$

Now that we have $n_p$ neurons for each of the $2d + 1$ neurons that we want to approximate. We can utilize them as ground set in solving a subset sum problem to achieve a small approximation error of each $w_j\phi(x_j)$.

Using Lemma 8 or Lemma 10, we are successful with our approximation up to error $\epsilon''$ with probability at least $1 - \delta''$ if

$$n_p \geq CM\log\left(M/\min\{\delta'', \epsilon''\}\right).$$

Putting everything together, we have to ensure that the pruning works (see Eq. (12)) with probability $(1 - \delta')$ and that every approximation of each of the $N$ non-zero parameters is successful with probability $(1 - \delta'')$. This is achieved with probability $(1 - \delta)$ for $\delta'' = \delta' = \delta/(N + 1)$, as we obtain from a union bound. Furthermore, we can allow an approximation error of each single parameter of $\epsilon' = \epsilon/Q$, as we derived earlier. This can all be achieved by

$$n_{1,0} \geq CMd\log\left(\frac{M}{\min\left\{\delta/(N+1), \epsilon/Q\right\}}\right).$$

**Case $L > 2$**:

In comparison to the case $L = 2$, we have the additional advantage that we can also prune Layers

$2, \ldots, L - 1$. As in the proof of the case $L = 2$, we need in total $n_{\text{tot}} = (2d + 1)n_p$ neurons for a successful approximation. But utilizing Corollary 11, we can distribute these $n_{\text{tot}}$ neurons on $L - 1$ layers. Thus, each layer has to cover only $n_{\text{tot}}/(L - 1)$ neurons.

$$n_{l,0} \geq C \frac{Md}{L-1} \log \left( \frac{M}{\min \{\delta'/(N + 1), \epsilon/Q\}} \right).$$

are therefore sufficient to for a successful approximation with probability at least $1 - \delta'$.

Our construction that achieves this is visualized in Figure 1 (e). It consists of $(L - 1)$ subset sum blocks, i.e., subnetworks with $\lceil (2d + 1)n_p/(L - 1) \rceil$ neurons in the intermediary layer and $m$ outputs. Each of these intermediary outputs, say $y_k$ (represented by a blue node in the figure), is connected to the final corresponding output $x_k^L$ along a path of neurons of the form $w_{L-1}\phi(...\phi(w_j y_k))$ with $w_j > 0$, thus adding $\prod_{j=1}^{L-2} w_j y_k$ to the final output $x_k^L$. Exemplary paths are also highlighted in orange in the figure.

Corollary 11 states that each of these contributions is valid in a subset sum approximation, as long as we can bound the path weight in such a way that we can integrate it into our universal constant of each width requirement. This is the case for $1 \leq \prod_{j=1}^{L-2} w_j \leq 1/\alpha$ for an $\alpha < 1$. In each layer, we need to find maximally $m + d$ of such $w_j$. Lemma 1 provides us with the result that a random neuron can be pruned to represent such a neuron $\phi(w_j x)$ on that path with probability at least $q$ so that $\alpha = 3/4$. Note that $q$ and $\alpha$ are independent of $L$. This guarantees with the application of a union bound that

$$n_{l,w} \geq C \log((m + d)/\delta'') \tag{13}$$

neurons need to be available for pruning to find the desired $2(m + d)$ neurons with probability at least $1 - \delta''$, where $C = \log(1/(1 - q))$.

Putting everything together, using a union bound, we can guarantee an overall existence probability of at least $1 - \delta$ by setting $\delta'' = \delta/(2(m + d)(L - 1) + N + 1)$ and $\delta' = \delta(N + 1)/(2(m + d)(L - 1) + N + 1)$, which leads to

$$n_{l,0} \geq C \frac{Md}{L-1} \log \left( M/ \min \{\delta/(2(m + d)(L - 1) + N + 1), \epsilon/Q\} \right).$$

$\square$

### B.2 Univariate lottery tickets

A similar reasoning as for linear multivariate lottery tickets allows us to prove our results for univariate lottery tickets, the second important building block of our representation of basis functions. However, we generally need 3 layers to implement a subset sum approximation instead of the 2 layers required in case of linear multivariate lottery tickets. Note that other results on network pruning would require at least two blocks and thus exactly 4 layers in the mother network (Fischer & Burkholz, 2021) (or could not represent the architecture because they are restricted to zero biases Pensia et al. (2020)). Our 3-layer block leverages the specific neural network structure that we use to approximate an univariate function, which we introduce next.

It is well known that every deep neural network with ReLU activation functions $\phi(x) = \max(x, 0)$ is a piecewise linear function (Arora et al., 2018). Conversely, each piecewise linear function with a finite number of linear regions can be represented by a ReLU network as long as it has enough neurons (Arora et al., 2018; Daubechies et al., 2019).

Deep neural networks are generally overparametrized. Yet, all parameters of an univariate network $f$ can be mapped to the effective parameter vectors $\boldsymbol{a}$ and $\boldsymbol{s}$ that represent $f$ in the following way. The knots $\boldsymbol{s} = (s_i)_{i \in [N-1]}$ mark the boundaries of the linear regions and $\boldsymbol{a} = (a_i)_{i \in [N]}$ indicate changes in slopes $m_i = (f(s_{i+1}) - f(s_i)) / (s_{i+1} - s_i)$ (with $s_N := s_{N-1} + \epsilon$) from one linear region to the next. Assuming $f(x) = a_N$ for $x < s_0$, we can write

$$f_N(x) = \sum_{i=0}^{N-1} a_i \phi(x - s_i) + a_N \tag{14}$$

with $a_i = m_i - m_{i-1}$ for $1 \leq i \leq N - 1$, $a_0 = m_0$, and $a_N = f(s_0)$.

The relevant domain, where $f$ varies non-linearly, is defined by $[s_0, s_{N-1}]$. Without loss of generality, we assume that $f$ is positive so that $f(x) \geq 0$ for all $x \in [s_0, s_{N-1}]$. This can always be achieved by an affine linear transformation of the output that is learned by the last layer. Alternatively, the parameters of the deep neural network could be transformed appropriately. The last layer of the randomly initialized mother network $f_0$ can have either ReLU or linear activation functions, since we assume $f(x) \geq 0$ so that $\phi(f(x)) = f(x)$ holds.

For a given $f$ of this form, we want to find a lottery ticket that is an $\epsilon$-approximation of $f$. The first question that we have to answer is how much error we are allowed to make in each parameter in order to achieve this.

### B.2.1 PRUNING ERROR FOR AN UNIVARIATE LOTTERY TICKET

**Lemma 13** (Approximation of univariate piecewise linear function). *Let $\epsilon > 0$, $f$ be defined as in Eq. (14) and $f_\epsilon$ be a piecewise linear function whose parameters fulfill $|a_i - a_{i,\epsilon}| \leq \epsilon_a$ and $|s_i - s_{i,\epsilon}| \leq \epsilon_s$ with*

$$\epsilon_a = \frac{\epsilon}{2\left((N+1)\max\{|s_0|, |s_{N-1}|\} + \sum_{i=0}^{N-1}|s_i|\right)}, \qquad \epsilon_s = \frac{\epsilon}{2\left(N + \sum_{i=0}^{N-1}|a_i|\right)}. \qquad (15)$$

*Then $\max_{x \in [s_0, s_{N-1}]} |f(x) - f_\epsilon(x)| \leq \epsilon$.*

*Proof.* It follows from the definition of $f$, the triangle inequality, and the fact that $|\phi(x) - \phi(y)| \leq |x - y|$ that

$$|f(x) - f_\epsilon(x)| \leq \sum_{i=0}^{N-1} |a_i - a_{i,\epsilon}||\phi(x - s_i)| + |a_{i,\epsilon}||\phi(x - s_i) - \phi(x - s_{i,\epsilon})| + |a_N - a_{N,\epsilon}|$$

$$\leq \sum_{i=0}^{N-1} \epsilon_a |x - s_i| + |a_{i,\epsilon}||s_i - s_{i,\epsilon}| + \epsilon_a$$

$$\leq \epsilon_a \left( N|x| + 1 + \sum_{i=1}^{N} |s_i| \right) + \sum_{i=0}^{N-1} |a_{i,\epsilon}|\epsilon_s$$

$$\leq \epsilon_a \left( N|x| + 1 + \sum_{i=1}^{N} |s_i| \right) + \sum_{i=0}^{N-1} (|a_i| + \epsilon_a)\epsilon_s$$

$$\leq \epsilon_a \left( N|x| + 1 + \sum_{i=1}^{N} |s_i| \right) + \sum_{i=0}^{N-1} (|a_i| + 1)\epsilon_s$$

$$\leq \epsilon_a \left( N\max\{|s_0|, |s_{N-1}|\} + 1 + \sum_{i=1}^{N} |s_i| \right) + \epsilon_s \left( N + \sum_{i=0}^{N-1} |a_i| \right)$$

$$\leq \frac{\epsilon}{2} + \frac{\epsilon}{2} < \epsilon$$

for all $x \in [s_0, s_{N-1}]$, where we have used Eq. (15) in the second to last inequality and assumed that $|s_{N-1}| \geq 1$. $\qquad \square$

Letting $Q = \max\{|s_0|, |s_{N-1}|\}$, $M = (1 + \max(\max_i |a_i|, \max_j |s_j|))$, and using similar arguments, we can more generally derive that

$$\max_{x \in S} |f(x) - f_\epsilon(x)| \leq \epsilon_a(1 + NQ + (N-1)M) + \epsilon_s NM \leq \epsilon_a N(Q + M) + \epsilon_s NM$$

where we used the fact that $M \geq 1$ in the last step. It is therefore sufficient to bound the error in each individual parameter by $|\theta_i - \theta_{i,\epsilon}| \leq \epsilon/(2N(Q + M))$.

For convenience, we state again Thm. 4 before we prove it.

**Statement** (Existence of univariate lottery ticket). *Assume that $\epsilon, \delta \in (0,1)$ and an univariate target network $f : \mathbb{R} \supset \mathbb{S}$ in form of Eq. (14) consisting of $N$ intermediary neurons are given. Define $M := (1 + \max(\max_i |a_i|, \max_j |s_j|))$ and $Q = \max_{x \in S} |x|$. Let each weight and bias of $f_0$ with depth $L \geq 2$ and architecture $\bar{n}_0$ be initialized according to Def. 3. Then, with probability at least $1 - \delta$, $f_0$ contains a sparse approximation $f_\epsilon \subset f_0$ so that $\max_{x \in [s_0, s_{N-1}]} |f(x) - \lambda f_\epsilon(x)| \leq \epsilon$ with*

$$n_{l,0} \geq C \frac{\max\{M, N\}}{L-2} \log \left( \frac{M}{\min\{\delta / [L(N+2) - 1], \epsilon/(2(Q+M))\}} \right). \tag{16}$$

*for $L \geq 3$ and*

$$n_{1,0} \geq C \frac{MQ}{\epsilon} \log \left( \frac{M}{\min\{\delta/(N+1), \epsilon/(2(Q+M))\}} \right).$$

*for $L = 2$, where the output is scaled by $\lambda = \prod_{l=1}^{L} (2\sigma_{w,l}^{-1})$.*

*Proof.* Let us start with the case $L = 3$.

**Case $L = 3$:**
Note that we can represent our target network $f(x) = \sum_{i=0}^{N-1} a_i \phi(x - s_i) + a_N$ as univariate neural network consisting of 4 layers

$$f(x) = \sum_{i=1}^{N} a_i \phi \left( \phi(x) - \phi(-x) - s_i \phi(1) \right) + \text{sign}(a_N) \phi(|a_N| \phi(1)),$$

where the first layer has width $n_1 = 3$ and the second layer has width $n_2 = N + 1$. (Note that we can also apply another ReLU activation function to the output, $\phi(f(x))$, because we assumed that $f(x)$ is positive. We omitted it for clarity.) Lemmas 8 or 10 let us solve the associated $(2N+1)$ subset sum problems if the first layer represents enough neurons of type $\phi(x)$ and $\phi(1)$ with

$$n_{1,0} \geq CM \log \left( \frac{M}{\min\{\delta/(3N+2), \epsilon/(2N(Q+M))\}} \right), \tag{17}$$

where each parameter $(1, -1, -s_i)$ is approximated up to error $\epsilon/(2N(Q+M))$. The $n_{1,0}$ neurons serve the creation of $n_{2,0}$ neurons in the next layer, which are approximately of the form $\phi(x - s_i)$ and $a_N$. These in turn are used to solve another set of subset sum problems that approximate the parameters $a_i$. This is feasible for

$$n_{2,0} \geq CN \log \left( \frac{3N+2}{\delta} \right). \tag{18}$$

**Case $L \geq 3$:**
Following the same steps of reasoning as in our proof of the existence of linear multivariate lottery tickets (see Figure 1), we can distribute the approximation of $N + 1$ neurons on $L - 2$ layers. The approximation of the target $f$ is thus successful with probability $1 - \delta$ if each of the $2N + 1$ subset sum problems is solved and we find the following number of connections (of type $w_j$) that create paths between our intermediary outputs and the final output. The first layer needs one connection 1, the second also 1, and all $L - 2$ remaining layers need maximally $N + 2$ connections. In total, this results in $\delta' = \delta/((N+2)L - 1)$ and thus

$$n_{l,0} \geq C \frac{\max\{M, N\}}{L-2} \log \left( \frac{M}{\min\{\delta / [L(N+2) - 1], \epsilon/(2(Q+M))\}} \right). \tag{19}$$

**Case $L = 2$:**
The case $L = 2$ is more complicated because we do not have enough layers to construct univariate neurons of the form $\phi(x)$, $\phi(-x)$, and $\phi(1)$. Instead we have to create the required $N$ neurons of the form $\phi(x - s)$ and $a_N \phi(1)$ directly in Layer 1. We can still achieve this approximation by utilizing Corollary 7 and thus solving $N + 1$ subset sum problems. Note that we can associate a neuron $w_2 \phi(w_1 x + b)$ in $f_0$ that is used to approximate a neuron $a_i \phi(x - s_i)$ of $f$ with a random variable $X$. We construct $X$ such that it contains a uniform distribution $U[-1, 1]$. With $n_{1,0}$ of such $X$, we can then approximate our target $f$ using Corollary 7.

Thus, let us derive the distribution of $X$. With probability at least $q$ we have that $1 \leq w_1 \leq 2$. ($q = 1/4$ for uniformly initialized $w_1$ and $q = 1/8$ for normally initialized $w_1$.) Given this $w_1$, we can approximate $s_i$ within error $\epsilon'$ with probability at least $\epsilon' w_1/2 \geq \epsilon' 1/2$ by pruning neurons to have appropriate $b$. $w_2$ contains a uniform distribution $U[-1, 1]$ with $\alpha \geq 1/4$ We can use this uniform distribution $U[-1, 1]$ to approximate $a_i/w_1$ up to error $\epsilon'/w_1 \leq \epsilon'/2$. All together, the random variable $X$ that reflects the described pruning contains a uniform distribution $U[-1, 1]$ with $\alpha = \epsilon'/8$. We can use $n_{1,0}$ random independent copies of $X$ to approximate each of the $N + 1$ neurons up to error $\epsilon'/2$ with probability at least $1 - \delta'$. According to Corollary 7, this is achieved for

$$n_{1,0} \geq CM \frac{1}{\alpha} \log \left( \frac{M}{\min \{\delta'/(N+1), \epsilon'\}} \right), \tag{20}$$

where the choice $\epsilon' = \epsilon/(2Q)$ and $\delta' = \delta$ proves the claim of the theorem. □

## C  EXISTENCE OF UNIVERSAL LOTTERY TICKETS

In the following, we present theorems about the construction of lottery tickets as concatenated basis functions. Both versions, one for polynomial and the other one for Fourier basis functions, share the general set-up so that we start with a joint proof beginning.

*Proof.* To show that $f_\epsilon$ is a strongly universal lottery ticket with respect to the function family $\mathcal{F}$, we have to prove that it fulfils Definition 1. Thus, for every $g \in \mathcal{F}$ we have to find $\boldsymbol{W} \in \mathbb{R}^{m \times k}$ and $\boldsymbol{c} \in \mathbb{R}^m$ so that

$$\sup_{x \in [0,1]^d} \|\boldsymbol{W} f_\epsilon(\boldsymbol{x}) + \boldsymbol{c} - g(\boldsymbol{x})\| \leq \epsilon. \tag{21}$$

Since $g \in \mathcal{F}$, we can write each component as linear combination with respect to our basis functions $b_i \in \mathcal{B}$ so that $g_i(\boldsymbol{x}) = \sum_{j=1}^k a_{ij} b_j(\boldsymbol{x}) + d_i$ with $a_{ij}, d_i \in [-1, 1]$. If we define $w_{ij} = a_{ij}\lambda$ and $c_i = d_i$ for a positive scaling factor $\lambda > 0$, we receive

$$\sup_{\boldsymbol{x} \in [0,1]^d} \|\boldsymbol{W} f_\epsilon(\boldsymbol{x}) + \boldsymbol{c} - g(\boldsymbol{x})\| \leq \sup_{\boldsymbol{x} \in [0,1]^d} \sum_{i=1}^m \sum_{j=1}^k |a_{ij}| |(\lambda f_{\epsilon,j}(\boldsymbol{x}) - b_j(\boldsymbol{x}))| \tag{22}$$

$$\leq km \sup_{\boldsymbol{x} \in [0,1]^d, j \in [k]} |\lambda f_{\epsilon,j}(\boldsymbol{x}) - b_j(\boldsymbol{x})| \tag{23}$$

using $|a_{ij}| \leq 1$ and the triangle inequality. Note that this inequality holds for any $p$-norm $\|\cdot\|$ with $p \geq 1$. Thus, $f_\epsilon$ is a strongly universal lottery ticket with respect to $\mathcal{F}$ if each component $j$ is bounded as

$$\sup_{\boldsymbol{x} \in [0,1]^d} |\lambda f_{\epsilon,j}(\boldsymbol{x}) - b_j(\boldsymbol{x})| \leq \frac{\epsilon}{km}. \tag{24}$$

To simplify the notation, in the remainder we drop the index $j$ but understand that both $f_\epsilon(\boldsymbol{x})$ and $b(\boldsymbol{x})$ have 1-dimensional output. To bound the error, we decompose it into two parts, one that captures the approximation of $b(\boldsymbol{x})$ by a target deep neural network $f_N$ (and thus a piece-wise linear function) and one that handles the error related to pruning a mother network $f_0$ to obtain the actual lottery ticket $f_\epsilon$.

$$\sup_{\boldsymbol{x} \in [0,1]^d} |\lambda f_\epsilon(\boldsymbol{x}) - b(\boldsymbol{x})| \leq \underbrace{\sup_{\boldsymbol{x} \in [0,1]^d} |\lambda f_\epsilon(\boldsymbol{x}) - f_N(\boldsymbol{x})|}_{\text{pruning}} + \underbrace{\sup_{\boldsymbol{x} \in [0,1]^d} |f_N(\boldsymbol{x}) - b(\boldsymbol{x})|}_{\text{approximation}} \tag{25}$$

$$\leq \frac{\epsilon}{2km} + \frac{\epsilon}{2km} = \frac{\epsilon}{km}.$$

The last inequality needs to be shown. The approximation error can be controlled by a sensible choice of the deep neural network architecture, while the pruning error is handled by Theorems 4 and Theorems 3. In both cases, we also have to consider, however, how the error propagates through the deep neural network layers. The following arguments depend on the specifics of the construction. □

## C.1   Polynomial universal lottery tickets

We continue with the proof for polynomial basis functions. For convenience, we restate Thm. 5 from the main manuscript.

**Statement.** *Let $\mathcal{B} \subset \left\{ \prod_{i=1}^{d} 0.5^{r_i}(1+x_i)^{r_i} \mid r_i \le q, x \in [0,1] \right\}$ be a subset of the polynomial basis functions with bounded maximal degree $q$ of cardinality $k = |\mathcal{B}|$. Let $\mathcal{F}(\mathcal{B})$ be the family of bounded (affine) linear combinations of these basis functions. For $\epsilon, \delta \in (0,1)$, with probability $1 - \delta$ up to error $\epsilon$, $f_0$ contains a strongly universal lottery ticket $f_\epsilon \subset f_0$ with respect to $\mathcal{F}$ if $f$ has $L_{log} \ge 2$ convolutional layers, followed by $L_{multi} \ge 2$ fully connected layers, and $L_{exp} \ge 2$ convolutional layers with channel size or width chosen as in Thms. 4 and 3 with the following set of parameters.*

*Logarithm: $\delta_{log} = 1 - (1-\delta)^{1/3}$, $\epsilon_{log} = \frac{\epsilon}{6tkm}$, $N_{log} = 1 + \left\lceil \sqrt{\frac{tkm}{\epsilon}} \right\rceil$, $M_{log} = 2$, $Q_{log} = 1$.*

*Multivariate: $\delta_{multi} = 1 - (1-\delta)^{1/3}$, $\epsilon_{multi} = \frac{\epsilon}{6km}$, $N_{multi} = kd$, $M_{multi} = q$, $Q_{multi} = d$.*

*Exponential: $\delta_{exp} = 1 - (1-\delta)^{1/3}$, $\epsilon_{exp} = \frac{\epsilon}{6km}$, $N_{exp} = 1 + \left\lceil t\sqrt{\frac{km}{4\epsilon}} \right\rceil$, $M_{exp} = 2$, $Q_{exp} = t \log 2$.*

*Proof.* Our objective is to bound the approximation and pruning error in Eq. (25) sufficiently.

**Approximation error**:
First, we bound the approximation error in Eq. (25) by $\epsilon/(2km)$. Our basis function $b(\boldsymbol{x})$ as well as our approximating deep neural network $f_N(\boldsymbol{x})$ can be written as composition of three functions. $b(\boldsymbol{x}) = g\left( \sum_{i=1}^{d} a_i h(x_i) \right)$, where $h(x) = \log((1+x)/2)$, $0 \le a_i \le q$, $\sum_{i=1}^{d} a_i \le t$, and $g(y) = \exp(y)$ for $-\log(2)t \le y \le 0$, $g(y) = 0.5^t$ for $y < -\log(2)t$, and $g(y) = 1$ for $y > 0$. $f_N(\boldsymbol{x}) = g_N\left( \sum_{i=1}^{d} a_i h_N(x_i) \right)$, where $g_N$ and $h_N$ are piece-wise linear approximations of $g$ and $h$ respectively. As we have restricted our approximation to a compact domain, all functions are Lipschitz continuous. Here, and in the sequel, we use the fact that on any interval, the slope of the piecewise linear function $g_N$ is bounded by that of $g$, which follows from the construction of $g_N$ in Eq. (14). Therefore, on any interval, the Lipschitz constant of $g$ is also the Lipschitz constant of $g_N$. Thus we can bound the difference between both functions as

$$
\begin{aligned}
\sup_{\boldsymbol{x} \in [0,1]^d} |f_N(\boldsymbol{x}) - b(\boldsymbol{x})| &= \sup_{\boldsymbol{x} \in [0,1]^d} \left| g\left( \sum_{i=1}^{d} a_i h(x_i) \right) - g_N\left( \sum_{i=1}^{d} a_i h_N(x_i) \right) \right| \\
&\le \sup_{\boldsymbol{x} \in [0,1]^d} \left| g\left( \sum_{i=1}^{d} a_i h(x_i) \right) - g\left( \sum_{i=1}^{d} a_i h_N(x_i) \right) \right| + \left| g\left( \sum_{i=1}^{d} a_i h_N(x_i) \right) - g_N\left( \sum_{i=1}^{d} a_i h_N(x_i) \right) \right| \\
&\le L_g t \sup_{x \in [0,1]} |h(x) - h_N(x)| + \sup_{x \in [-t\log(2)-\epsilon', \epsilon']} |g(x) - g_N(x)|,
\end{aligned}
\tag{26}
$$

where we bounded each $a_i$ by $q$ and $\epsilon'$ denotes the pruning error for now. $L_g$ denotes the Lipschitz constant of $g(y)$ and $g_N(y)$ on their domain $[-t \log 2 - \epsilon', \epsilon']$, which is bounded by 1.

Thus, bounding

$$
\sup_{x \in [0,1]} |h(x) - h_N(x)| \le \frac{\epsilon}{4kmt} \quad \text{and} \quad \sup_{x \in [-t\log(2)-\epsilon', \epsilon']} |g(x) - g_N(x)| \le \frac{\epsilon}{4km} \tag{27}
$$

would be sufficient to control our approximation error. To achieve this, we have to allow for large enough number of neurons $N_{\text{exp}}$ and $N_{\text{log}}$ in the univariate representation (14) of $g_N$ and $h_N$.

**Approximation of $\mathbf{h(x) = \log((1+x)/2)}$**
Let us determine $N_{\text{log}}$ first. We partition the domain $[0,1]$ of $h_N(x)$ into intervals $I_i = [s_i, s_{i+1}) \subset [0,1]$ and $I_{N-1} = [s_{N-2}, s_{N-1}] \subset [0,1]$ of length $\Delta = 1/(N-1)$ based on equidistant knots $s_i$ with $s_0 = 0$ and $s_{N-1} = 1$. For every $x$ let $i_x$ mark the index of the interval in which $x \in I_{i_x}$ is contained. Since $h(x) = \log((1+x)/2)$ is concave, we have for all $x$

$$
|h(x) - h_N(x)| = h(x) - h_N(x) = h(x) - h(s_{i_x}) - (x - s_{i_x})\left( h(s_{i_x+1}) - h(s_{i_x}) \right)/\Delta, \tag{28}
$$

where $i_x := \arg\min_{i,\, x - s_i \geq 0} x - s_i$ is the index the corresponds to the interval that contains $x$. Furthermore, the maximum approximation error is attained on the interval $I_0 = [0, \Delta]$. Accordingly, we have

$$\sup_{x \in [0,1]} |h(x) - h_N(x)| = \sup_{x \in [0,\Delta]} \log(0.5(1 + x)) - \log(0.5) - x * m = \sup_{x \in [0,\Delta]} \log(1 + x) - x * m$$

with $m = (\log(0.5(1 + \Delta)) - \log(0.5))/\Delta = \log(1 + \Delta)/\Delta$. To identify the position of maximum error, we set the first derivative with respect to $x$ of the error objective to zero and obtain $x^* = 1/m - 1$. This results in

$$\sup_{x \in [0,1]} |h(x) - h_N(x)| = \sup_{x \in [0,\Delta]} \log(1 + x) - x * m = \log(1 + 1/m - 1) - (1/m - 1) * m$$

$$= -\log(m) + m - 1 \approx 1/2\Delta - 1/8\Delta^2 + 1/24\Delta^3 + 1 - 1/2\Delta$$

$$+ 1/3\Delta^2 - 1 \leq 1/4\Delta^2,$$

where we used the series expansion of $\log(1 + x)$. For $0.25\Delta^2 \leq \frac{\epsilon}{4kmt}$ we can therefore guaranty a small enough approximation error. Hence, the choice $N_{\log} = 1 + \left\lceil \sqrt{\frac{kmt}{\epsilon}} \right\rceil$ is sufficient to achieve the desired bound Eq. (27).

**Approximation of $\mathbf{g(x)} = \exp(\mathbf{x})$**
We can make a similar argument for $g(x) = \exp(x)$ on its domain $\mathcal{D} = [-t \log(2) - \epsilon', \epsilon']$. Since we cut off $g$ and $g_N$ outside of the actual support $[-t \log(2), 0]$ to avoid amplifying errors that we know are errors, we can just focus on the domain $[-t \log(2), 0]$. Note that also $\exp(\cdot)$ is monotonously increasing, but since it is convex, we have

$$\sup_{x \in [-t\log(2),0]} |g(x) - g_N(x)| = \sup_{x \in [-\Delta,0]} g_N(x) - g(x) = \sup_{x \in [-\Delta,0]} \exp(-\Delta) + m(x + \Delta) - \exp(x)$$

with $m = (1 - \exp(-\Delta))/\Delta \approx (\Delta - 0.5\Delta^2)/\Delta = 1 - 0.5\Delta$. The maximum is attained at $x^* = \log(m)$ so that we have

$$\sup_{x \in [-t\log(2),0]} |g(x) - g_N(x)| = \exp(-\Delta) + m(\log(m) + \Delta) - m \approx 1/8\Delta^2,$$

where we employed series expansions of $\exp(x)$ and $\log(1 + x)$. Again, from $\Delta = \frac{\log(2)t}{N_{\exp} - 1}$ follows that any $N_{\exp} \geq 1 + \left\lceil t\sqrt{\frac{km}{4\epsilon}} \right\rceil$ would be sufficient to achieve our approximation objective (27).

Last but not least we mention also the number of non-zero parameters that we want to prune in case of the multivariate function. As we can represent this function exactly by a deep neural network, we do not inflict any approximation error and we can set $N_{\text{multi}} = dk$ to the maximum number of its non-zero parameters.

With this, we have derived the stated $N$. We still have to bound the pruning error to conclude our proof.

**Pruning error**:
The pruning error in Eq. (25) has to be smaller or equal to $\epsilon/(2km)$, which we can achieve by limiting the pruning approximation errors $\epsilon_{\exp}$, $\epsilon_{\text{multi}}$, and $\epsilon_{\log}$ in Theorems 4 and 3 to represent the targets $g_N$, $h_N$ and the multivariate linear map with matrix $A$.

Similarly to Eq. (26), we have to consider the error propagation through a composition of functions. Yet, this time also the multivariate linear map inflicts an error. Recall that our deep neural network $f_N(\boldsymbol{x})$ can be written as composition of three functions so that $f_N(\boldsymbol{x}) = g_N\left(\sum_{i=1}^{d} a_i h_N(x_i)\right)$. Analogously, also our lottery ticket is of similar form: $f_\epsilon(\boldsymbol{x}) = g_\epsilon\left(a_\epsilon\left(h_\epsilon(x_1), ..., h_\epsilon(x_d)\right)\right)$, where $a_\epsilon$ denotes a function from $\mathbb{R}^d$ to $\mathbb{R}$. A slight complication is that we can approximate each lottery ticket only up to a scaling factor $\lambda$. The scaling factor of each function is not completely arbitrary because all together have to harmonize with the initialization of biases, as previously explained. The

exact scaling factors in the theorems ensure that with $\lambda = \lambda_{\exp}\lambda_{\text{multi}}\lambda_{\log}$, we can write

$$
\begin{aligned}
\sup_{\boldsymbol{x}\in[0,1]^d} |f_N(\boldsymbol{x}) - \lambda f_\epsilon(\boldsymbol{x})| &= \sup_{\boldsymbol{x}\in[0,1]^d} \left| g_N\left(\sum_{i=1}^d a_i h_N(x_i)\right) - \lambda_{\exp}g_\epsilon\left(\lambda_{\text{multi}}a_\epsilon\left(\lambda_{\log}\boldsymbol{h}_\epsilon(\boldsymbol{x})\right)\right) \right| \\
&\leq \sup_{\boldsymbol{x}\in[0,1]^d} \left| g_N\left(\sum_{i=1}^d a_i h_N(x_i)\right) - g_N\left(\lambda_{\text{multi}}a_\epsilon\left[\lambda_{\log}\boldsymbol{h}_\epsilon(\boldsymbol{x})\right]\right) \right| \\
&+ \sup_{\boldsymbol{x}\in[0,1]^d} \left| g_N\left(\lambda_{\text{multi}}a_\epsilon\left[\lambda_{\log}\boldsymbol{h}_\epsilon(\boldsymbol{x})\right]\right) - \lambda_{\exp}g_\epsilon\left(\lambda_{\text{multi}}a_\epsilon\left[\lambda_{\log}\boldsymbol{h}_\epsilon(\boldsymbol{x})\right]\right) \right| \\
&\leq \sup_{\boldsymbol{x}\in[0,1]^d} L_{g_N}\left| \sum_{i=1}^d a_i h_N(x_i) - \lambda_{\text{multi}}a_\epsilon\left[\lambda_{\log}\boldsymbol{h}_\epsilon(\boldsymbol{x})\right] \right| + \underbrace{\sup_{y\in[-qd\log(2)-\epsilon',\epsilon']} |g_N(y) - \lambda_{\exp}g_\epsilon(y)|}_{\leq \epsilon_{\exp}} \\
&\leq \sup_{\boldsymbol{x}\in[0,1]^d} L_{g_N}\left| \sum_{i=1}^d a_i\left(h_N(x_i) - \lambda_{\log}h_\epsilon(x_i)\right) \right| + \underbrace{\sup_{\boldsymbol{x}\in[0,1]^d} L_{g_N}\left| \sum_{i=1}^d a_i\lambda_{\log}h_\epsilon(x_i) - \lambda_{\text{multi}}a_\epsilon\left[\lambda_{\log}\boldsymbol{h}_\epsilon(\boldsymbol{x})\right] \right|}_{\leq \epsilon_{\text{multi}}} + \epsilon_{\exp} \\
&\leq \underbrace{\sup_{\boldsymbol{x}\in[0,1]} L_{g_N}t\left|h_N(x) - \lambda_{\log}h_\epsilon(x)\right|}_{\leq \epsilon_{\log}} + L_{g_N}\epsilon_{\text{multi}} + \epsilon_{\exp} \\
&\leq L_{g_N}t\epsilon_{\log} + L_{g_N}\epsilon_{\text{multi}} + \epsilon_{\exp},
\end{aligned}
\tag{29}
$$

where we have used Theorems 4 and 3 to bound the difference between a target network and the corresponding lottery ticket. Theorem 3 depends on $M = q$ (since all parameters are bounded by $q$) and $Q = d$, since $h(x) \in [0, 1]$ and the inputs to the multivariate linear function are bounded this way. Since $L_{g_N} \leq 1$, we can thus achieve a pruning error of maximally $\epsilon/(2km)$ with with $\epsilon_{\log} = \epsilon/(6kmt)$, $\epsilon_{\text{multi}} = \epsilon/(6km)$, and $\epsilon_{\exp} = \epsilon/(6km)$.

With the stated parameter choice, each of the three parts of the lottery ticket exists independently with probability at least $(1 - \delta)^{1/3}$. All three exist thus with probability at least $\left((1 - \delta)^{1/3}\right)^3 = (1 - \delta)$, which concludes our proof. $\qquad\square$

### C.1.1 Fourier universal lottery tickets

Similarly, we can also bound the approximation and pruning error in case of the second family of functions that we have considered. In doing so, we prove the following theorem.

**Theorem 14.** *For $\epsilon, \delta \in (0, 1)$, with probability $1 - \delta$ up to error $\epsilon$, $f_0$ with the specified properties contains a strongly universal lottery ticket $f_\epsilon \subset f_0$ with respect to the function family $\mathcal{F}(\mathcal{B})$, which is defined as bounded affine linear combinations with respect to a finite subset of the Fourier basis functions $\mathcal{B} \subset \left\{\sin\left(2\pi\left(\sum_{i=0}^d n_i x_i + c\right)\right) \mid n_i \in [M](\forall i \in [d]), c \in [0, 1], x_i \in [0, 1]\right\}$ with cardinality $k = |\mathcal{B}|$. $f_0$ consists of $L_{multi} \geq 2$ fully connected layers followed by $L_{sin} \geq 2$ convolutional layers whose width or number of channels fulfill the requirements of Thms. 4 and 3 with the following set of parameters.*
*Multivariate: $\delta_{multi} = 1 - (1 - \delta)^{1/2}$, $\epsilon_{multi} = \frac{\epsilon}{8km\pi}$, $N_{multi} = dk$, $M_{multi} = 1 + M$, $Q_{multi} = d$.*
*Sin: $\delta_{sin} = 1 - (1 - \delta)^{1/2}$, $\epsilon_{sin} = \frac{\epsilon}{4km}$, $N_{sin} = 1 + \left\lceil \pi\arcsin\left(\epsilon_{sin}/2\right)^{-1} \right\rceil$, $M_{sin} = 1 + (d + 1)M$, $Q_{sin} = (d + 1)M$.*

*Proof.* As for polynomial lottery tickets, we have to bound the approximation error and the pruning error separately to be smaller or equal to $\epsilon/(2km)$ to fulfill Eq. 25.

**Approximation error**:
The basis function $b(\boldsymbol{x})$ as well as our approximating deep neural network $f_N(\boldsymbol{x})$ can be written as composition of two functions. $b(\boldsymbol{x}) = g\left(\sum_{i=1}^d n_i x_i + c_i\right)$, where $g(y) = \sin(2\pi y)$, $0 \leq n_i \leq M$,

and $c \in [0, 1]$. Correspondingly, $f_N(\boldsymbol{x}) = g_N \left( \sum_{i=1}^d n_i x_i + c_i \right)$, where $g_N$ is a piece-wise linear approximations of $g$. As we have restricted our approximation to a compact domain, all functions are Lipschitz continuous and we can bound the difference between both functions as

$$
\begin{aligned}
\sup_{\boldsymbol{x} \in [0,1]^d} |b(\boldsymbol{x}) - f_N(\boldsymbol{x})| &= \sup_{\boldsymbol{x} \in [0,1]^d} \left| g \left( \sum_{i=1}^d n_i x_i + c_i \right) - g_N \left( \sum_{i=1}^d n_i x_i + c_i \right) \right| \\
&\leq \sup_{y \in [0, dM+1]} |g(y) - g_N(y)| = \sup_{y \in [0, dM+1]} |\sin(2\pi y) - g_N(y)|,
\end{aligned}
$$

where $[0, dM + 1]$ is the domain of $g$ and $g_N$. As in the previous proof, we assume that $g_N$ approximates $g$ on an equidistant grid with linear regions of size $\Delta = 1/(N_{\sin} - 1)$. Similarly as before, a piece-wise linear approximation inflicts the maximal error

$$
\begin{aligned}
\sup_{y \in [0, dM+1]} |\sin(2\pi y) - g_N(y)| &\leq 2 \sup_{z,y \in [0, dM+1-\Delta] \text{ s.t. } |z-y| \leq \Delta} |\sin(2\pi z) - \sin(2\pi y)| \\
&= 4 \sup_{z,y \in [0, dM+1-\Delta] \text{ s.t. } |z-y| \leq \Delta} |\sin(\pi(z-y)) \cos(\pi(z+y))| \\
&\leq 4 \sup_{x \in [0,\Delta]} |\sin(\pi x)| = 4 \sin(\pi \Delta)
\end{aligned}
$$

using the addition theorem $\sin(\alpha + \beta) - \sin(\alpha - \beta) = 2 \sin(\beta) \cos(\alpha)$ with $\alpha = \pi(z + y)$ and $\beta = \pi(z - y)$ for the first equality and assuming $\Delta \leq 1/2$ to obtain the last equality.

It follows that $N_{\sin} \geq 1 + \left\lceil \pi \left( \arcsin \left( \epsilon/(8km) \right) \right)^{-1} \right\rceil$ leads to sufficiently small approximation error. Note that $N_{\sin}$ is always bigger than 3 so that $\Delta \leq 1/2$ is fulfilled automatically.

**Pruning error**:
The argument for the pruning error is quite similar but the error related to the linear transformation $a_N(\boldsymbol{x}) = \sum_{i=1}^d a_i x_i + c_i$ is non-zero and needs to be controlled as well. With $\lambda = \lambda_{\sin} \lambda_{\text{multi}}$ we can derive

$$
\begin{aligned}
\sup_{\boldsymbol{x} \in [0,1]^d} |f_N(\boldsymbol{x}) - \lambda f_\epsilon(\boldsymbol{x})| &= \sup_{\boldsymbol{x} \in [0,1]^d} |g_N(a_N(\boldsymbol{x})) - \lambda_{\sin} g_\epsilon(\lambda_{\text{multi}} a_\epsilon(\boldsymbol{x}))| \\
&\leq \sup_{\boldsymbol{x} \in [0,1]^d} |g_N(a_N(\boldsymbol{x})) - g_N(\lambda_{\text{multi}} a_\epsilon(\boldsymbol{x}))| + \sup_{\boldsymbol{x} \in [0,1]^d} |g_N(\lambda_{\text{multi}} a_\epsilon(\boldsymbol{x})) - \lambda_{\sin} g_\epsilon(\lambda_{\text{multi}} a_\epsilon(\boldsymbol{x}))| \\
&\leq L_{g_N} \underbrace{\sup_{\boldsymbol{x} \in [0,1]^d} |a_N(\boldsymbol{x}) - \lambda_{\text{multi}} a_\epsilon(\boldsymbol{x})|}_{\leq \epsilon_{\text{multi}}} + \underbrace{\sup_{y \in [0, dM+1]} |g_N(y) - \lambda_{\sin} g_\epsilon(y)|}_{\leq \epsilon_{\sin}}
\end{aligned}
$$

$$
\leq L_{g_N} \epsilon_{\text{multi}} + \epsilon_{\sin} \leq \epsilon/(2km)
$$

for $\epsilon_{\text{multi}} = \epsilon/(8km\pi)$ and $\epsilon_{\sin} = \epsilon/(4km)$, where we have used that $L_{g_N} = 2\pi$, i.e., the Lipschitz constant of the function $\sin(2\pi \cdot)$.

These estimates hold in reference to Theorems 4 and 3.

It might seem strange that these errors do not depend on $M$. However, in the end, the widths of both mother neural networks does depend on $M$, as Theorems 4 and 3 consider the domain and maximum value of the parameters of the functions $g_N$ and $a_N$ in the error assessment.

We conclude that a lottery ticket and those both parts of the neural network exist with probability at least $(1 - \delta)^{1/2}(1 - \delta)^{1/2} = 1 - \delta$. □

