# OpenReview forum: "On the Existence of Universal Lottery Tickets"
_ICLR.cc/2022/Conference — ICLR 2022 Poster_

### Official Review · Reviewer_y2DU · 2021-11-02

**Correctness:** 4
**Technical Novelty And Significance:** 2
**Empirical Novelty And Significance:** Not applicable
**Recommendation:** 6
**Confidence:** 3

**Main Review:**

Overall, I think the paper is well written and studies an interesting direction. However, my main concern is the significance of the result.

As shown in Theorem 1, the number of neurons is linearly proportional to the number of base functions |B|. Such linear seems to make the result a bit trivial than expected. Specifically, I feel the theory is essentially doing the following thing:
1. Given a set of base functions, for each function, we prune a randomly initialized network for approximation.
2. We then concatenate all the randomly pruned networks together (allowing no interaction between those networks in each layer), which gives the universal lottery ticket network.

I didn’t dig into the details of the proof but I think following the above strategy we will end up with the same rate shown in this paper with a linear dependency on k.
Overall, it seems this paper utilizes the above way for construction with additional effort to deal with the scaling issue, which is not difficult.

**Summary Of The Paper:**

This paper studies the existence of universal lottery tickets, a stronger version of lottery tickets that allows transfer learning.
The key idea is to show that, by pruning a network, we are able to approximate a finite set of universal base functions.


**Summary Of The Review:**

Overall, I think this paper studies an interesting direction. However, due to some concerns on the significance of the result, I am not sure whether the result is above the acceptance bar of ICLR. I currently lean to give a weak rejection but I am happy to adjust the score based on the authors' response.

---

> ### Author Response · Authors · 2021-11-21
> **We do not prune monomials independently**
>
> We thank Reviewer y2DU for taking the time to review our paper and would like to highlight how and why our results are non-trivial.
> 1. In fact, we do not prune a randomly initialized neural network to approximate each base function separately. On the contrary, we combine convolutional and fully connected layers to avoid exactly that. Firstly, we use convolutional layers to apply the same univariate function to each component. (We thus do not waste parameters doing that.) Secondly, combine univariate functions by multivariate linear layers in a single layer.
> For instance, Yarotsky (2017) and Hieber (2020) would need to multiply polynomials pairwise and require a considerable number of layers to do that.
> Their approach, which can leverage an arbitrary amount of depth and can thus get away with very little width, would need a high number of layers of width $6 (1+dq)^d d$ or at least $dk$, which also scales with $k$. Our construction scales much better than that except for the fact that it depends on 1/sqrt{\epsilon}. (An \epsilon dependence is necessary if we want to have a constant depth).
> 2. We improved our dependence to $\sqrt{k}$. However, note that the final feature layer needs a width of $k$, of course.
> 3. We would like to take this opportunity to further highlight the significance of our results. We have
> formalized a notion of universal lottery tickets;
> introduced a novel sparse representation of polynomial and Fourier basis functions as composition of univariate functions and linear transformations, which benefits from parameter sharing in convolutional layers;
> derived improved LT existence results of independent interest based on a novel technique to distribute subset sum blocks between different layers.

---

> > ### Comment · Reviewer_y2DU · 2021-12-05
> > **Thanks for the updated result.**
> >
> > Thanks authors for the updated result.
> >
> > In my previous review comments, I am not saying that you are constructing your subnetwork using the naive approach I mentioned, but if the rate does not improve over the naive construction, the significance seems low.
> >
> > The updated result addresses my concerns and hence I raise my score 1.

---

### Official Review · Reviewer_TK28 · 2021-11-03

**Correctness:** 3
**Technical Novelty And Significance:** 3
**Empirical Novelty And Significance:** 3
**Recommendation:** 6
**Confidence:** 3

**Main Review:**

Overall this paper is technically sound and well organized. Besides, my comments are as follows:

---

By looking at Theorem 1, it seems that the major contribution compared to existing works (in terms of the theoretical results) is that this paper does not require the mother network to have depth $2L$ to approximate the target network with depth $L$, but could be of any depth, at the price of using more neurons per-layer. Could you elaborate on this? It would be better to clarify this technical difference in the introduction.

---

Besides, theorem 1 states that the universal approximation is taken with respect to the function family $F(B)$. But as claimed by the authors, functions in $F(B)$ are not exactly the target neural network function $f$. There seemly has a gap since the lottery ticket hypothesis states that a larger network with pruning can approximate a smaller network. Perhaps the authors state this approximation results with respect to the target network somewhere, but it could be much better if you can directly highlight that as a theorem or corollary.

---

Could you also specifically show the case when the mother network has a smaller depth in the main section? This case is rather interesting to me since existing works cannot cover it.

---

I do not quite get the illustration in Figure 1. It seems that the authors want to show how to develop an architecture to approximate a monomial. However, as stated on page 4, g(x) and h(x) are exponential and logarithmic functions, which I am not clear how to use FC-net and Conv-net to approximate (where we only have linear transformation and ReLU activation).


**Summary Of The Paper:**

This paper develops a theory to explain the lottery ticket hypothesis based on extensions of subset-sum results and a strategy to leverage higher amounts of depth. The developed proof could be of independent interest since it leverages the power of depth and improves the existing bounds.

**Summary Of The Review:**

The authors may need to carefully address my concerns and questions, or I am not sure whether this paper is of good quality or sufficient importance.

---

> ### Author Response · Authors · 2021-11-21
> **Clarification of main ideas and universal lottery ticket construction**
>
> We thank Reviewer TK28 for the questions, which we are happy answer. To clarify the points, we have also revised our draft and included our main theorems on polynomial lottery tickets in the main manuscript.
> 1. We have added in the introduction that, in the general setting when we want to prune a mother network with depth $L_0$ to approximate a target network with depth $L_t$ and width $n_t$, we require the mother network to have a width $n_0 \geq O(n_t L_t/L_0 \log\left(n_t L_0/(L_t\epsilon) \right))$.
> We still need the mother network to have a larger depth than the target network. We could leverage the case in which the mother network has a higher depth than twice the target depth, i.e., $L_0 \geq 2 L_t$.
> This is particularly useful in the case that is the main focus of our paper, when we specifically construct target network representations (which are universal) that have small depths.
> 2. We agree with Reviewer TK28 and state the main approximation theorems for polynomials in our revised draft. Specifically, we can approximate $\log(0.5(1+x))$ or $\exp(x)$ up to error $\epsilon$ with an univariate 2-layer neural network of width $O(1/ /sqrt{\epsilon}).
> The linear multivariate network can be represented exactly.
> Next, we approximate these target networks by pruning a mother network with the width as stated in the new Theorem 5.
> 3. Unfortunately, we cannot cover the case when the target network has a larger depth than the mother network. However, we can deal with a case when the mother network has a depth that is smaller than twice the depth of the target network. Because of the space constraint, we were not able to add the proof idea for that case in the main manuscript but we would like to invite Reviewer TK28 to take a look at the Case L=2 on page 20.
> The main idea is to create neurons of the form $\phi(x-s)$ directly in layer 1 and create subset sum blocks with these neurons that depend on two parameters rather than 1. This is more difficult because this requires the neurons of the form $\phi(w1 x + b)$ to have a similar s = -b/w1. As it is less likely that a neuron his this property, we need to have a wider layer to ensure that we find enough of them.
> 4. Figure 1 is supposed to explain the construction of the full universal lottery ticket. Let us focus on the construction of polynomials. Each blue column in the first layers approximates the function $\log(0.5(1+x))$. In fact, exactly the same function is applied to every input component $x_i$, which can be achieved with convolutional layers of filter dimension 1. We just need to parameterize a network that approximates $\log(0.5(1+x))$ ones. If we had only fully connected layers, we would need to do that for each input component independently and up with d times the number of parameters than we need if we use convolutional layers.
> This way we have constructed a vector of the form $v \approx [\log(0.5(1+x_1)), …, \log(0.5(1+x_d)) ]$. Next we linearly transform this vector to receive a vector $y$ with $k$ components. For instance, the first component looks like $y_1 = \sum^k_{i=1} a_{1i} v_i $.
> Next, we use again convolutional layers to apply a function that approximates $\exp(y_i)$ to every component of the vector $y$. The result is a vector $b(x)$. Each of its components approximates a monomial, for instance, the monomial $\prod^d_{i=1} 0.5^{a_{1i}}(1+x_i)^{a_{1i}}$.

---

> > ### Comment · Reviewer_TK28 · 2021-12-08
> > **Thanks for your response**
> >
> > Thanks for providing a detailed response to my questions. Now I understand why it is sufficient to use log function in Figure 1 since you have already shown that a two-layer network can approximate this function. I also understand that the main improvement is make the requirement on the width of a large network be dependent on the network depth. I would like to raise my evaluation to 6.

---

### Official Review · Reviewer_6yJz · 2021-11-04

**Correctness:** 4
**Technical Novelty And Significance:** 3
**Empirical Novelty And Significance:** 2
**Recommendation:** 8
**Confidence:** 4

**Main Review:**

As written in the summary above, this paper provides some novel results and intuitive ideas to show the existence of universal lottery tickets - lottery tickets which can be slightly tuned to approximate any function in a function class. However, there are two main concerns that I have:

1. Looking at the theorem statements in the appendix, the parameter $N$ in Theorem 1 might be of the order $dk/\epsilon$. In that case, the required width $n_{l,0}$ becomes super linear in $d$ and $k$, and no-longer remains logarithmic in $\epsilon$. This would be bad, particularly since modern ML applications have huge dimension $d$.

2. The paper only talks about approximating the functions in the basis $\mathcal{B}$, whereas the more interesting case is approximating classes of functions such as Lipschitz-continuous functions. However, I think, applying this paper's results directly for that setting would lead to the classic 'curse of dimensionality', where the basis would need to contain $k=(h(1/\epsilon))^d$ functions, where $h$ is some monotonically increasing function. While this might be unavoidable, this does put the point of universal lottery tickets into question, that is, can there exist practical, small-sized universal lottery tickets in high dimensions?

NOTE: I might be missing something important here in the two points above, so the authors should feel free to correct me.

Other than this, I think the main paper (first 9 pages) is not able to explain a lot of important aspects and I had to go through the appendix to understand them. One suggestion could be to just focus on one of the bases (say polynomial or fourier) in the main paper, and defer others to the appendix. This would give the readers a more direct understanding of how the construction works.

**Summary Of The Paper:**

This paper proves that large enough randomly initialized ReLU networks can contain subnetworks, which can act as universal lottery tickets, that is, only by tuning the last layer of these subnetworks, they can approximate any (reasonably well-behaved) function. The paper does this by showing that large enough randomly initialized ReLU networks contain subnetworks such that their last hidden layer can act as the basis for a class of functions, and hence the output layer can simply take a linear combination of the neurons of the last hidden layer to approximate any function in that class. The paper proves this for polynomial and Fourier basis. Another key contribution of this paper is that it demonstrates that large depth can be leveraged to reduce the width needed for the strong lottery ticket hypothesis to hold.

**Summary Of The Review:**

The paper provides some novel results and intuitive ideas to show the existence of universal lottery tickets, and shows that depth can be used to reduce the width required for strong lottery ticket hypothesis. However, I have some concerns regarding the required width of the network, which might limit the overall practicality of the results.

---

> ### Author Response · Authors · 2021-11-21
> **Improvements on width requirements**
>
> We thank Reviewer 6yJz for the positive assessment of our work and the insightful points of critique.
> 1. This is true. Unfortunately, other constructions of polynomials have to accept a similar width requirement. Even the arbitrary depth construction by Yarotsky (2017) and Hieber (2020) requires a constant width of $6 (1+dq)^d d$ (see Lemma A.4) and almost all of the $O(log(1/\epsilon))$ many layers have this width. (In some cases this could be reduced to $6 kd$ though.) In contrast, we have a constant depth and are now able to reduce N to $O(d\sqrt{k/\epsilon})$. Furthermore, this width is only required in one layer (or L-2 layers with width $O(d/(L-2)\sqrt{k/\epsilon})$).
> Note that our idea to use convolutional layers eliminates already a number of problems related to $d$. In fact, our width depends on the maximum degree $t$ of the monomials, thus, we have $N = O(t\sqrt{k/\epsilon})$. This degree $t$ could be smaller than $d$ in applications. We have updated our theorems with this more precise factor.
> While $d$ can indeed be huge in many machine learning applications, many problems have effectively a much smaller dimension. If this is the case, the first neural network layers could linearly (or non-linearly) transform the input into such a smaller dimensional space, as also proposed by Malach et al. (2020). However, the obtained lottery tickets could then only be universal with respect to functions that vary on this smaller dimensional space.
> 2. We thank Reviewer 6yJz for making this connection to general approximation results on Lipschitz continuous functions. Indeed, we could not overcome the curse of dimensionality and have added a discussion to our main manuscript. Yarotsky (2018) has shown that a feed forward neural network needs $O(\omega_f(O(\epsilon^{-d/2}))$ parameters to represent a function with modulus of continuity $\omega_f$. Therefore, sparser universal lottery tickets cannot exist at least if they are subnetworks of fully-connected neural networks. This is a great motivation why we limited ourselves to k-sparse function families.
> 3. We have revised our draft and included our main theorems on universal polynomial lottery tickets in the main paper. We thank Reviewer 6yJz for the suggestion.

---

> > ### Comment · Reviewer_6yJz · 2021-12-06
> > **Reply to author response.**
> >
> > I thank the authors for addressing my concerns. I suggest the authors should add the discussions above to their paper. I am maintaining my score.

---

### Official Review · Reviewer_rD3v · 2021-11-04

**Correctness:** 3
**Technical Novelty And Significance:** 2
**Empirical Novelty And Significance:** Not applicable
**Recommendation:** 6
**Confidence:** 2

**Main Review:**

I think this paper provides some interesting ideas on how to study the universal lottery ticket hypothesis. However, I think the paper may have some significant limitations.

The most important issue I find in this paper is that the results are not rigorous enough and the sparsity level of the lottery ticket is not addressed properly. This paper only has an informal theorem in the main paper, and has more detailed results in the appendix. However, after going over them, it is still not very clear to me what exactly is the sparsity level of the universal lottery ticket that is proven to exist. This is very important, as without guarantees on sufficient sparsity, one can just set the mask $B$ in Definition 2 to be all ones to use the original network as a lottery ticket of itself. Moreover, the authors set a sparsity level of 0.5 in the experiments. I feel that this may not be sparse enough. When the sparsity is just at a constant level, it seems that we can simply randomly delete a constant proportion of the neurons in each layer of the network, and then the universality of the lottery ticket can still be proved based on the universality of random feature models.

Theorem 1 is also a bit confusing. According to Definition 1, if $\lambda f_{\epsilon}$ is strongly universal, isn’t $f_{\epsilon}$ also strongly universal? If this is the case, it seems pointless to introduce $\lambda$.

The result of this paper is also a bit incremental, as it is mainly a combination of known results on using polynomial/Fourier bases to approximate smooth functions and using neural networks to approximate polynomial/Fourier bases.

In terms of the presentation of the results, the authors may consider presenting formal theorems in the main paper, and add discussions to justify the results.

**Summary Of The Paper:**

This paper gives proof of the existence of universal lottery tickets. Specifically, a definition of universal lottery tickets is proposed, and then proved based on functions approximation results for polynomials, Fourier series using neural networks.


**Summary Of The Review:**

For the reasons listed above, I think the current paper needs some further improvement. Therefore I would like to recommend rejection.

---

> ### Author Response · Authors · 2021-11-21
> **We discuss sparsity in depth in our revised manuscript.**
>
> We thank Reviewer rD3v for the constructive feedback.
> 1) Our results are fully rigorous. We prove all our theorems and lemmas. We even prove extensions of the original subset sum approximation results. In the revised draft, we have stated the main theorems in full rigor and refrained from a high level summary (which would be easier to digest). However, please note that also before all our theorems were stated in full rigor in the appendix.
> 2) Sparsity: We have to distinguish two possible definitions of sparsity. First, we could be interested in the total number of non-zero parameters that are used by the lottery ticket. This definition would be in line with the literature on neural network expressivity, which we discuss now in detail in our revised draft. The common definition in the lottery ticket community is to measure the sparsity of a lottery ticket relative to the mother network. According to this definition, it is trivial to achieve extremely sparse tickets, because one could simply add more neurons to the mother network, which are not needed and can be pruned away fully. For that reason, the objective from a theoretical point of view is to provide an as small as possible bound on the number of required parameters of the mother network.
> To combine these two seemingly contradictory objectives, we would like to find an as sparse as possible target network representation and then minimize the number of required parameters in the mother network. We believe we are the first to bring these two objectives together.
> A big advantage of our approach is furthermore that we construct tickets explicitly and we could indeed report the number of non-zero parameters in our construction. We have therefore added this information in our experiments and thank Reviewer rD3v for the suggestion.
> Sparsity in experiments: Note that only the edge-popup algorithm finds LTs at sparsity 0.5, which is a known limitation of the algorithm (see original paper) but not of our proofs.
> When we prune following our proof strategy, we find tickets of sparsity of 0.02 and less, which is sparser than what most pruning algorithms can find in practice. Furthermore, as explained above, we could always make our tickets sparser relative to the mother network if we start from a bigger mother network. We have added a measurement of sparsity to the experiment section.
> 3) We have proposed a non-standard representation of polynomials and Fourier basis functions that utilizes convolutional layers to obtain sparser representations. In our revised draft, we discuss the differences between our approach and the literature in detail.
> 4) Note that we indeed mentioned in our revised draft that \lambda could be integrated into the weights of the linear regression for universal lottery tickets. However, our statement on lottery ticket pruning is more general and applies also to non-universal functions but then requires the consideration of \lambda. We have distinguished these two use cases more clearly in our revised draft.
> 5) In addition, we would like to highlight the further novelty of our contributions, as we have
> formalized a notion of universal lottery tickets;
> introduced a novel sparse representation of polynomial and Fourier basis functions as composition of univariate functions and linear transformations, which benefits from parameter sharing in convolutional layers;
> derived improved LT existence results based on a novel technique to distribute subset sum blocks between different layers.

---

> > ### Comment · Reviewer_rD3v · 2021-12-07
> > **Thanks for your clarification**
> >
> > I appreciate the authors' response to my original review. Most of my major concerns have been addressed fairly well. Therefore I decide to increase my score.

---

### Decision · Program_Chairs · 2022-01-20

**Decision:**

Accept (Poster)

**Comment:**

Dear Authors,

The paper was received nicely and discussed during the rebuttal period. The current consensus suggests the paper be accepted, but could have another round of revisions before it gets published

- Definition of sparsity within theoretical results + clarity of results. This seems to be the main concern by one of the reviewers.
- The reviewers acknowledged that some of the concerns raised could be found somewhere in the appendix, which raises further the concern of the presentation of the results: reviewers suggest a more focused and proper dissemination of the results (main theorems in main text + explanation of the results obtained, etc), which requires another round of revisions and reviewing.

Best AC